

# Weather regimes and related atmospheric composition at a Pyrenean observatory characterized by hierarchical clustering of a 5-year data set

Jérémy Gueffier[1], François Gheusi[1], Marie Lothon[1], Véronique Pont[1], Alban Philibert[1], Fabienne Lohou[1], Solène Derrien[1], Yannick Bezombes[1], Gilles Athier[1], Yves Meyerfeld[1], and Antoine Vial[1]

[1]LAERO, CNRS, Université Toulouse 3 Paul Sabatier, France

**Correspondence:** Jérémy Gueffier (jeremy.gueffier@aero.obs-mip.fr) & François Gheusi (francois.gheusi@aero.obs-mip.fr)

**Abstract.** Atmospheric composition measurements taken at many high-altitude stations around the world, aim to collect data representative of the free troposphere and of an intercontinental scale. However, the high-altitude environment favours vertical mixing and the transportation of air masses at local or regional scales, which has a potential influence on the composition of the sampled air masses. Mixing processes, source-receptor pathways, and atmospheric chemistry may strongly depend on

local and regional weather regimes, and these should be characterized specifically for each station. The Pic du Midi (PDM) is a mountaintop observatory (2850 m a.s.l.) on the north side of the Pyrenees. PDM is associated with the Centre de Recherches Atmosphériques (CRA), a site in the foothills ar 600 m a.s.l. 28 km north-east of the PDM. The two centers make up the Pyrenean Platform for the Observation of the Atmosphere (P2OA). Data measured at PDM and CRA were combined to form a 5-year hourly dataset of 23 meteorological variables notably: temperature, humidity, cloud cover, wind at several altitudes. The

dataset was classified using hierarchical clustering, with the aim of grouping together the days which had similar meteorological characteristics. To complete the clustering, we computed several diagnostic tools, in order to provide additional information and study specific phenomena (foehn, precipitation, atmospheric vertical structure, and thermally driven circulations). This classification resulted in six clusters: three highly populated clusters which correspond to the most frequent meteorological conditions (fair weather, mixed weather and disturbed weather, respectively); a small cluster evidencing clear characteristics of

winter northwesterly windstorms; and two small clusters characteristic of south foehn (south to southwesterly large-scale flow, associated with warm and dry downslope flow on the lee side of the chain). The diagnostic tools applied to the six clusters provided results in line with the conclusions tentatively drawn from 23 meteorological variables. This, to some extent, validates the approach of hierarchical clustering of local data to distinguish weather regimes. Then statistics of atmospheric composition at PDM were analysed and discussed for each cluster. Radon measurements, notably, revealed that the regional background

in the lower troposphere dominates the influence of diurnal thermal flows when daily averaged concentrations are considered. Differences between clusters were demonstrated by the anomalies of $CO$, $CO_2$, $CH_4$, $O_3$ and aerosol number concentration, and interpretations in relation with chemical sinks and sources were proposed.





## 1 Introduction

The Pic-du-Midi (PDM) is a high peak, situated at 2877 m above sea level and located to the north of the main watershed of the
Pyrenean chain (white line in Fig. 1) and dominating the French plain. A scientific observatory was established on the summit in
1878, and since then it has been a key location for atmospheric observations in the Pyrenees (Bücher and Dessens, 1991, 1995).
For almost three decades, it has worked jointly with the Centre de Recherches Atmosphériques (CRA), an experimentation site
in the foothills at 600 m a.s.l., near Lannemezan, 28 kilometres away (Fig. 1). These two sites form the Pyrenean Platform for
Observation of the Atmosphere (P2OA, https://p2oa.aeris-data.fr, described in Lothon et al. (2023)), operated by the University
of Toulouse 3 Paul Sabatier and CNRS INSU.

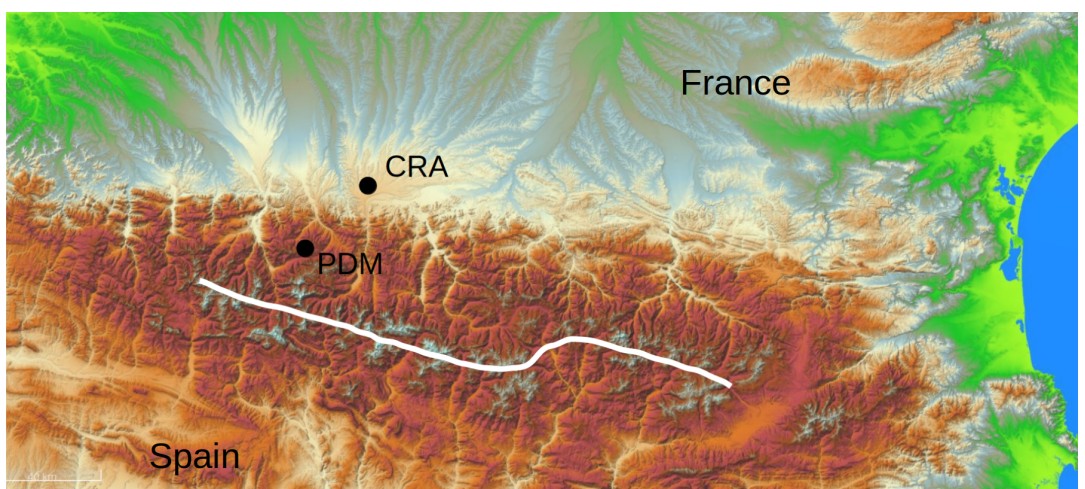

**Figure 1.** Location of measurement sites. The white line represents the main Pyrenean watershed.

Historical measurement series at PDM have been studied for temperature (Bücher and Dessens, 1991, 1995), relative humidity and cloud cover (Bücher and Dessens, 1995), rainfall (Bücher and Dessens, 1997) and ozone (Marenco et al., 1994).
The latter authors showed that ozone trends measured at the PDM throughout the twentieth century were in line with data from
other European high altitudes sites. This suggested the capacity of mountain observatories to provide atmospheric composi-
tion measurements representative of a vast geographic area, at least when long time periods are considered. In the same vein,
Chevalier et al. (2007) found close agreement between multi-year averaged ozone data from high altitude stations in Europe,
including PDM, and from airborne measurements in the free troposphere at the same altitude. Henne et al. (2010) compared
many European air quality observatories by means of a particle dispersion model, and found that PDM and a few other sites
for the most part are remote from anthropogenic emissions. Mountain sites thus appear to be suitable sites to provide baseline
concentrations representative of the free troposphere, remote from local sources (Parrish et al., 2014), and in turn representative
of the global scale (Keeling et al., 1976).

Nevertheless, mountain observatories, even those found on the peaks, lie in the mountain boundary layer and are not 100%
of the time representative of the regional free troposphere. At high-altitudes in the mountians, turbulence and airflows are





created and this may drag surface air masses to the mountaintops and then affect the gas and aerosols concentrations (Serafin
et al., 2018; Henne et al., 2004; Griffiths et al., 2014). At the Alpine station Jungfraujoch (3600 m a.s.l.), and transposable at
least to PDM, and likely also to other high-mountain stations around the world, Griffiths et al. (2014) points out the potential
influence of local to regional sources on air composition measurement using three main processes combining vertical transport
of boundary-layer air and mixing with background air during the recent history of the air mass. These processes are: (i)
thermally driven mountain boundary layers and anabatic flows; (ii) terrain-forced flows such as foehn, in which synoptic
winds significantly interact with the topography; (iii) deep vertical mixing over the surrounding plains followed by horizontal
advection to the mountains.

These three types of processes are relevant for the P2OA. Thermally induced circulations are generated by differential
heating of slope vs. valley atmospheres, or mountain vs. plain atmospheres. This results in anabatic (upward) flows during
the day, and conversely, katabatic (downward) flows at night, in a variety of spatial scales: slope flows, valley flows, and
plain-mountain flows (Whiteman, 2000). Particular focus on thermally driven circulations and their impact on atmospheric
composition has been given for many mountain sites (e.g Lugauer and Winkler, 2005; Necki et al., 2003; Forrer et al., 2000,
among many others). At the P2OA specifically, such studies have been conducted by Gheusi et al. (2011), Jiménez and Cuxart
(2014), Tsamalis et al. (2014), Román-Cascón et al. (2019), and Hulin et al. (2019). All these studies reveal a significant diurnal
impact of thermally driven circulations on atmospheric composition, and more specifically at PDM: a daytime enhancement
of any atmospheric species that are generally more concentrated in the boundary layer than in the free troposphere (e.g. water
vapor, radon, CO, $CH_4$, etc.); and conversely, a daytime depletion for species less concentrated in the (rural) boundary layer
than in the free troposphere (typically ozone, and $CO_2$ during the growing season). This suggests the partial or total influence
of the regional boundary layer at PDM in the daytime. Studying a case of anabatic transport from CRA to PDM with a simple
numerical transport model constrained by ozone measurements, Tsamalis et al. (2014) found a possible range of 14 to 57%
of boundary-layer air mixed into free-tropospheric air in the daytime. Hulin et al. (2019) tested three methods for detecting
thermally induced circulations in the P2OA area, which will be used later in the present study (Sections 2.3.2 and 4.3).

A second phenomenon which occurs frequenlty at the P2OA is foehn, a hydraulic-like flow pattern occurring on the lee-side
of a mountain barrier when the synoptic flow is forced to flow over and plunge beyond the crest line (Whiteman, 2000). In
the P2OA configuration (on the northern side of the Pyrennes, Fig. 1), foehn is a warm, dry, strong downslope southerly wind
affecting the northern flank of the Pyrenees. In case of deep foehns, the downslope flow may even reach the surface at CRA in
the foothills. Southerly foehn cases in the Pyrenes where studied during PYREX, which was a field campaign in the Pyrenees
regarding clear-air turbulence (Bougeault et al., 1997). However, to our knowledge, no climatology of south foehn is available
for the Pyrenees. The impact of foehn on ozone has been studied at an Alpine Italian foothill station subject to north foehn
winds (Weber and Prévôt, 2002) with results showing strongly reduced $O_3$ levels during foehn events in summer and slightly
increased levels in summer. At the Jungfraujoch, Forrer et al. (2000) confirm those ozone variations and show a strong increase
of CO and $NO_x$ concentrations.

The third phenomenon is vertical mixing in the plain due to deep convective systems followed by advection towards the
PDM. This transport occurs at a synoptic scale. Forrer et al. (2000) and Zellweger et al. (2003) thoroughly studied the impact



on, respectively, CO and NO$_y$ concentrations showing a similar impact on concentrations than the two other phenomena. At
P2OA however, deep convection often occurs over the mountain, notably in the case of unstable southwesterly or westerly
flows, and advected toward the plain for example at CRA.

Another common way to analyse air composition data at mountain sites is to track (usually at a larger continental scale) the
air masses using backward trajectories or dispersion models. The composition measurements are then sorted by geographical
source regions (e.g., Cui et al. (2011) and Loeoev et al. (2008) at the Jungfraujoch; Cristofanelli et al. (2013) at Campo
Imperatore, central Italy; Perry et al. (1999) at Mauna Loa, Hawaii; among many). The two studies at the Jungfraujoch highlight
differences in ozone concentration and other chemical components such as CH$_4$ and CO (Loeoev et al., 2008), depending on
whether air masses were influenced by the European planetary boundary limit or were long-range advected to Jungfraujoch.
In Cristofanelli et al. (2013) differences of ozone concentrations are shown at Campo Imperatore, in the Abruzzi massif,
depending on whether air masses come from the Mediterranean sea or from the European continent. Continental air masses
tend to bring air masses with higher ozone values.

A survey of local or regional weather regimes also brings useful information, since meteorology may affect atmospheric
composition at mountain sites in many, and often complex ways, through different transport and mixing patterns at all scales,
contrasting conditions for photochemistry or atmospheric scavenging by precipitation, etc. Thus, sorting composition data by
weather types may be a fruitful approach. A rich variety of methods to build meteorological classifications is encountered in the
literature. Weather regimes may be computed from pressure fields using global weather models, and the resulting classification
is generally intended for large geographical areas, e.g. Europe (Cortesi et al., 2019), or the Mediterranean basin (Giuntoli et al.,
2021). At smaller scales, studies aiming at characterizing weather regimes at specific measurement sites exist for urban areas
(e.g. Hidalgo and Jougla (2018) for Toulouse, France; Hodgson and Phillips (2021) for Birmingham, United Kingdom). In
the latter study, the authors use local meteorological data and an algorithm of hierarchical clustering to build a meteorological
classification. For the Alps, a classification of weather types has been in existence for a long time since 1945 by Schüepp
(1979) and described in Stefanicki et al. (1998). This synoptic weather type classification system (SYNALP) is determined
by 4 parameters (speed of surface geostrophic wind, direction and speed of the 500 hPa wind, height of the 500 hPa surface
and baroclinicity) measured or computed for a circular area (diameter 444 km) covering Switzerland. Stefanicki et al. (1998)
studied the frequency of changes in those weather types since 1945 and showed an increase of convective days in winter at the
expense of advective days. In addition, in Coen et al. (2011), SYNALP was utilized to analyse a long time-series of chemical
species (including aerosols and CO) measured at the Jungfraujoch, to assess the influence of free tropospheric air and air
advected from the boundary layer to the Jungfraujoch.

However, no such meteorological classification exists for the Pyrenean area. Our main objective in this study is to provide
a classification of observation days at the P2OA sorted by typical synoptic weather regimes, and to establish statistics for all
variables – especially gas and particle concentrations – in the different regimes. We chose to use only data produced locally on
the platform, and to use hierarchical clustering as the classification method. This approach has the advantage of being easily
applicable to other observatories by local investigators who have easy access to data measured in situ. Also, large-scale model



fields may miss local meteorological specificities (due to small-scale topography, field heterogeneity, etc.) that are otherwise captured by in situ measurements.

Hierarchical clustering allows us to obtain weather pattern data without any preconceptions about the local weather. It is a classification method that groups data vectors (in the multi-dimensional space of all considered variables) depending on their closeness (details given in Section 2.2). Carried out on a dataset of meteorological variables, it will generate clusters with similar meteorological characteristics. Such clusters can be linked to weather regimes, and hierarchical clustering has thus been widely utilized for this goal (e.g., Kalkstein et al., 1987; Ng et al., 2020; Hodgson and Phillips, 2021, among many other

references). With the aim of characterizing the specific impact at the P2OA of the synoptic meteorological context, the day-to-day changes are relevant to drive the clustering, but not the seasonal variations (e.g., of temperature) nor the variations related to the diurnal thermal cycle. Multi-year trends present for some variables (e.g., $CO_2$) are not in our scope either. Therefore, a pre-processing will be applied to the raw observation data to filter out those undesirable components and keep the day-to-day variations only.

Details of the measurements and the database, the data processing, the clustering method and the diagnostic tools are provided in Section 2. Meteorological regimes obtained from the clustering are presented in Section 3, and compared with diagnostic tools designed to focus on specific meteorological phenomena (Section 4). Finally we will consider and compare the statistics of the atmospheric composition variables available at PDM in the different meteorological clusters (Section 5). Conclusions are drawn, and perspectives suggested, in the final Section 6.

## 2    Methods

### 2.1    Dataset

#### 2.1.1    Database and hourly dataset

The time frame of the present study runs from the 1st of January 2015 to the 31st of December 2019. The period was chosen to optimize data coverage for a panel of atmospheric measurements at both P2OA sites. All instruments used in our study, with

technical details and the output variables are listed in Table 1. The entire P2OA dataset is described in Lothon et al. (2023).

We adopted the Coordinated Universal Time (UTC) for the whole study, because this time standard is almost the same as local solar time, since PDM is at longitude $0.14°E$).

As the data was collected at different time resolutions (Table 1), a first dataset was built on a synchronized hourly basis. Thus, the values provided for a given timestamp (e.g., 27 May 2018, 08:00:00 UTC) represent averages of any data available

in the one-hour interval beginning at this timestamp (08:00:00-08:59:59). Even though the hierarchical clustering detailed below in Section 2.2 is based on a final daily data set, building an intermediary hourly data set was needed for some of the diagnostic tools (Section 2.3) working on an hourly basis – for example, the detection of diurnal cycles of wind or chemical concentrations.




We used an extensive set of meteorological data recorded routinely at the PDM (2877 m a.s.l.) and the CRA (600 m a.s.l.)
stations (data available online at https://p2oa.aeris-data.fr/) which included temperature, relative humidity and pressure mea-
sured by standard weather stations, cloud occurrence above the CRA and wind measured at different levels above the ground
up to the mid-troposphere, as detailed here. At CRA, a 60 m tower provides meteorological measurements at both low and
fast frequency. We considered the measurements of mean wind from this tower (10m) and from two wind profilers. They all
provided the three components of the wind, but over different vertical ranges and resolutions. The UHF (1274 MHz) wind
profiler scans the lower troposphere between 100 m up to 6 km above the ground, with a vertical resolution of 75 m. The
VHF (45 MHz) wind profiler covers the range 1.5-16 km agl in the mid troposphere to the lower stratosphere, with a 375-m
resolution. Technical details are available in Campistron et al. (1999) for the VHF and in Jacoby-Koaly et al. (2002) for the
UHF. At PDM, we considered the standard measurements of wind at 2 m above the ground (note that surface wind at PDM is
affected by buildings in some wind sectors – Hulin et al., 2019).

The ground-based anemometers and the two wind profilers at CRA provided wind time series for many vertical levels.
However, wind data at two close levels (e.g., 100 m and 200 m above the ground) are strongly correlated and provide redundant
information. Thus, we selected key vertical levels, in a sufficient amount to capture the vertical structure of the dynamics from
the ground up to the mid-troposphere, but not too many in order to avoid redundancy. Therefore we chose surface levels at
PDM (2877 m a.s.l.) and CRA (600 m a.s.l. + 10 m a.g.l. = 610 m a.s.l.), and higher levels at 750 m a.s.l., 1600 m a.s.l. and
2850 m a.s.l. above CRA.

Additional observation data were also considered and are listed in Table 1. For the consideration of cloud cover, a full-
sky imager was used to retrieve a cloud-cover fraction by means of the algorithm ELIFAN (Lothon et al., 2019) based on
the red-over-blue ratio and a blue sky library. We used the rain gauge at CRA for precipitation estimationss. Finally, surface
energy and sensible and latent heat flux, deduced from the high-rate measurements at 30 m, were considered, to take account
of surface/atmosphere interactions.

To study the impact on the atmospheric composition measured at PDM, we considered the measurements of atmospheric
composition ($CH_4$, $CO_2$, CO, $O_3$) and aerosol particle numbers ((Hulin et al., 2019, and references therein for details on the
instruments). Note that the CO data series used here is a composite of data from two instruments: an IR absorption analyser
and a cavity-ringdown spectrometer, Table1). These were needed in order to upgrade the data coverage over the studied period
at a satisfactory level. Radon volume activity has also been included in the present dataset even though the radon monitor has
only been in operation only since October 2017.

### 2.1.2 Daily dataset suitable for separating synoptic weather regimes

As discussed in the introduction (Section 1), the focus of this study is on the synoptic weather regimes at the P2OA and their
impact on observations. The typical timescale for synoptic weather changes is a few days. Therefore, from the basic hourly
dataset detailed in the former paragraph, we built a final daily data set composed of 1826 days, and where (i) the diurnal
variations have been neutralized by averaging on a daily base; (ii) multi-annual and seasonal trends were characterized by
means of a non-linear least-square regression, then removed.



**Table 1.** Instrumentation characteristics

| Instrument | Start date | Type | Time resolution | Site | Variable (unit) |
|---|---|---|---|---|---|
| **Main meteorological variables** | | | | | |
| Automatic Weather Station | | | | | |
| (Vaisala Inc., QMH 102 sensor) | 06/2004 | In situ | 5 minutes | PDM | Temperature (K) and relative humidity (%) |
| (Vaisala PMT16A sensor) | 06/2004 | In situ | 5 minutes | PDM | Pressure (hPa) |
| Campbell HMP45 | 05/2011 | In situ | 10s | CRA | Temperature (K) |
| Campbell HMP45 | 01/2011 | In situ | 1 seconde | CRA | Relative humidity (%) |
| Barometer Vaisala PTB101B | 07/2010 | In situ | 10s | CRA | Pressure (hPa) |
| UHF Wind profiler | 04/2010 | Remote | 5 minutes | CRA | Wind components $u, v, w$ (m s$^{-1}$) |
| VHF Wind Profiler | 06/2001 | Remote | 15 minutes | CRA | Wind components $u, v, w$ (m s$^{-1}$) |
| **Additional meteorological variables** | | | | | |
| RAPACE sky imagery system | 06/2006 | Remote | 15 minutes | CRA | Cloud cover fraction (%) |
| CNR1 Kipp Zonen | 07/2010 | In situ | 1s | CRA | Radiative compenents (IR and visible) (W m$^{-2}$) |
| Rain Gauge ARG100 | 04/2011 | In situ | 10s | CRA | Rainfall (mm) |
| **High frequency meteorological measurements for energy flux estimates (30m)** | | | | | |
| Campbell CSAT3 sonic anemometer | 06/2010 | In situ | 0.1s | CRA | Wind components $u, v, w$ (m s$^{-1}$) & temperature (K) |
| Campbell Licor7500A | 06/2010 | In situ | 0.1s | CRA | Water vapour content (g m$^{-3}$) |
| **Atmospheric composition variables** | | | | | |
| UV Analyzer Thermo 49i | 04/1999 | In situ | 5 minutes | PDM | Ozone mole fraction (nmol mol$^{-1}$) |
| Radon monitor ANSTO 1500L | 10/2017 | In situ | 30 minutes | PDM | Radon volumic activity (mBq m$^{-3}$) |
| Cavity ring-down spectroscopy | 05/2014 | In situ | 4 secondes | PDM | $CO_2$ mole fraction ($\mu$mol mol$^{-1}$) |
| analyser (Picarro Inc. G2401) | | | | | $CH_4$ mole fraction (nmol mol$^{-1}$) |
| | | | | | CO mole fraction (nmol mol$^{-1}$) |
| IR analyser Thermo 48i | 01/2005 | In situ | 5 minutes | PDM | CO mole fraction (nmol mol$^{-1}$) |
| Condensation particle counter | 07/2008 | In situ | 5 minutes | PDM | Total suspended particle concentration (# cm$^{-3}$) |
| (TSI Inc 3010) | | | | | |

The regression function contains a linear part in order to model the long-term variation, and a 1-year periodic part with sinusoidal components (up to the fourth harmonic) to model the seasonal variability. The generic regression function thus writes for any variable $X$:

$$X(t) = a_1 + a_2 t + a_3 \sin(2\pi t) + a_4 \cos(2\pi t) + a_5 \sin(4\pi t) + a_6 \cos(4\pi t) + a_7 \sin(6\pi t) + a_8 \cos(6\pi t) + a_9 \sin(8\pi t) + a_{10} \cos(8\pi t)$$

($t$ being expressed in year).

After subtracting the long-term and seasonal trends from the daily averages, we obtained what we call anomalies in the rest
of the article. For example, in Fig. 2(a), we can see the original $CO_2$ time series of daily averages, as well as the modelled




long-term and seasonal trends. Figure 2(b) displays the resulting $CO_2$ anomaly. For the four irradiance variables, we divided the trends from the daily averages because the subtraction did not neutralize the seasonal variability.

The same process was applied for all variables tagged with "Anomaly" in Table 2 – the other variables being simple daily averages. The 23 variables which drive the clustering are listed in the upper part of Table 2. All other variables are used to

conduct statistical analyses within the obtained clusters but do not influence the clustering.

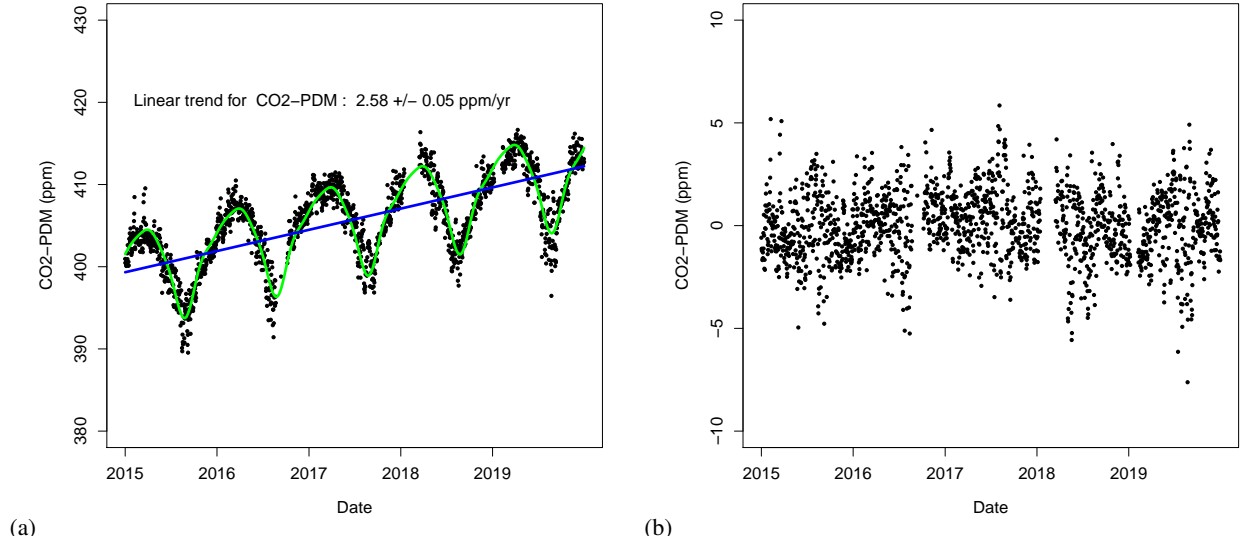

(a)                                                                                      (b)

**Figure 2.** Time series of $CO_2$ measured at the PDM for our time frame. In panel (a), the blue line represents the long-term trend of $CO_2$ and the green line represents the seasonal trend. Both are computed by nonlinear least-square regression. Panel (b) shows the obtained $CO_2$ anomaly time series.

## 2.2 Hierarchical clustering

Hierarchical clustering is a non-supervised classification method which builds groups of points that are the closest in the multi-dimensional space of all considered variables. In this space, a point – or event – represents, in our case, a vector formed by all variables of the daily data set associated with a given date. Thus, events falling in each cluster tend to share common

characteristics.

The key requirement in hierarchical clustering is the ability to assess distances in this space. Hierarchical clustering is an iterative method where, at each step, the closest points, or groups of points (clusters), are progressively merged into a new cluster (Wilks, 2011).

Over and above the way distances between two points are computed (Euclidean distance is usually used, and was adopted

in the present study), hierarchical clustering methods differ in the way distances between clusters are assessed. There are three



**Table 2.** List of variables in the daily dataset.

| Input Variables driving the clustering | Unit | Site | Vertical level | Anomaly ? |
|---|---|---|---|---|
| Temperature | K | CRA | 2m a.g.l. (Surface) | Yes |
| Temperature | K | PDM | 2m a.g.l. (Surface) | Yes |
| Relative Humidity | % | CRA | 45m a.g.l. (Surface) | Yes |
| Relative Humidity | % | PDM | 2m a.g.l. (Surface) | Yes |
| Specific Humidity | $g\,kg^{-1}$ | CRA | 45m a.g.l. (Surface) | Yes |
| Specific Humidity | $g\,kg^{-1}$ | PDM | 2m a.g.l. (Surface) | Yes |
| Pressure | hPa | PDM | 2m a.g.l. (Surface) | Yes |
| Cloud Cover | % | CRA | | No |
| Upward shortwave irradiance | $W\,m^{-2}$ | CRA | 60m a.g.l. (Surface) | Yes |
| Downward shortwave irradiance | $W\,m^{-2}$ | CRA | 60m a.g.l. (Surface) | Yes |
| Upward longwave irradiance | $W\,m^{-2}$ | CRA | 60m a.g.l. (Surface) | Yes |
| Downward longwave irradiance | $W\,m^{-2}$ | CRA | 60m a.g.l. (Surface) | Yes |
| Wind $u$ (west-east) component | $m\,s^{-1}$ | CRA | 10m a.g.l. (Surface) | No |
| Wind $v$ (south-north) component | $m\,s^{-1}$ | CRA | 10m a.g.l. (Surface) | No |
| Wind $u$ component | $m\,s^{-1}$ | PDM | 10m a.g.l. (Surface) | No |
| Wind $v$ component | $m\,s^{-1}$ | PDM | 10m a.g.l. (Surface) | No |
| Wind $u$ component (UHF profiler) | $m\,s^{-1}$ | CRA | 750 m a.s.l. | No |
| Wind $v$ component (UHF profiler) | $m\,s^{-1}$ | CRA | 750 m a.s.l. | No |
| Wind $u$ component (UHF profiler) | $m\,s^{-1}$ | CRA | 1600 m a.s.l. | No |
| Wind $v$ component (UHF profiler) | $m\,s^{-1}$ | CRA | 1600 m a.s.l. | No |
| Wind $u$ component (VHF profiler) | $m\,s^{-1}$ | CRA | 2850 m a.s.l. | No |
| Wind $v$ component (VHF profiler) | $m\,s^{-1}$ | CRA | 2850 m a.s.l. | No |
| Wind $w$ component (VHF profiler) | $m\,s^{-1}$ | CRA | 2850 m a.s.l. | No |

| Chemical variables measured at the PDM | | | | |
|---|---|---|---|---|
| Ozone mole fraction | $nmol\,mol^{-1}$ | PDM | 2m a.g.l. (Surface) | Yes |
| Carbon Dioxide mole fraction | $\mu mol\,mol^{-1}$ | PDM | 2m a.g.l. (Surface) | Yes |
| Carbon Monoxide mole fraction | $nmol\,mol^{-1}$ | PDM | 2m a.g.l. (Surface) | Yes |
| Methane mole fraction | $nmol\,mol^{-1}$ | PDM | 2m a.g.l. (Surface) | Yes |
| Particle numbers concentration | $\#\,cm^{-3}$ | PDM | 2m a.g.l. (Surface) | Yes |
| Radon volumic activity | $mBq\,m^{-3}$ | PDM | 2m a.g.l. (Surface) | No |



main methods: Ward's method (based on the sum of squarred errors between the two clusters, centroid (distance between the centroids of the clusters) and linkage (complete, single, or average). In the complete and single linkage, the maximum and minimum distances between points from the two groups are retained, respectively, whereas average linkage evaluates the average cluster-to-cluster distance (Wilks, 2011).

Ward's method is often used in meteorological studies due to its ability to form groups with balanced populations (Kalkstein et al., 1987). Nevertheless, those methods were applied on a meteorological data set and compared in Kalkstein et al. (1987). In their study, average linkage (with Euclidean distance) turned out to be the most suitable method as it minimized variance within clusters, compared to Ward's and centroid methods.

For this reason, we chose the linkage method for our study as we needed minimum variance within the cluster (well-defined weather regimes). With our dataset, however, single and average linkage resulted in one very large group and many single-day groups. Only the complete linkage method provided groups with more balanced populations, and hence this method was chosen for this study.

Meteorological studies using hierarchical clustering use a different approach than in ours. They tend to use a Principal Component Analysis (PCA) on their input dataset. This analysis counteracts the inter-dependence of their input variables. This inter-dependence exists when studies focus on a zone with different measurement sites with few variables (e.g. Degaetano (1996), Bravo et al. (2012), Pineda-Martínez and Carbajal (2017)). The PCA is also needed in studies that include rainfall in the clustering due to its shape (e.g. Hodgson and Phillips (2021), Ramos (2001), Ng et al. (2020)). We did not use a PCA in our study. However to avoid redundancy problems we carefully chose the input variables as described in the previous section. In addition, before the clustering, we centered and scaled our dataset.

## 2.3 Diagnostic tools

What we call diagnostic tools in this study are additional indicators computed mostly from data of our hourly or daily datasets, but in some cases also from additional observations (rain gauge) or an extra data source (NCEP meteorological model). Our main motivation is to further analyse the groups emerging from the hierarchical clustering, with a focus on specific atmospheric properties (e.g., vertical structure) or phenomena (e.g., foehn).

All the diagnostic tools are summarized in Table 3. They have been computed either on an hourly or daily basis (depending on the need), and separated into three thematic groups: atmospheric vertical structure, thermally driven circulations, and foehn effect.

### 2.3.1 Atmospheric vertical structure and precipitation

The first two diagnostic tools described below concern the vertical structure of the lower troposphere. Potential temperatures at both CRA and PDM sites have been computed from surface temperatures and pressures, so that the mean daily difference ($\Delta\theta = \theta_{\mathrm{PDM}} - \theta_{\mathrm{CRA}}$) between stations gives us an approximate but simple indication of the stability of the lower atmosphere in the area.



Another key variable is the daytime convective boundary layer (CBL) depth ($Z_i$), which is the depth over which any scalar may be mixed by convection in a short time range, generally less than one hour (Stull, 1988). This variable can be estimated

hourly at CRA with the UHF wind profiler. Here we use estimations from Philibert (2023), based on the fact that the turbulent CBL is topped by a temperature and moisture inversion, and a drop in turbulence. They are thus deduced from the local maximum reflectivity in the low troposphere, inversely weighted by the intensity of turbulence, as well as by criteria on temporal and spatial continuity. Sensible and latent heat fluxes were also considered, in order to take into account surface/atmosphere interactions and relate them to the observed CBL depth. We computed anomalies of those fluxes in the same way as other

variables as described earlier.

Completing moisture and cloud cover measurements, rainfall data are relevant but complex to handle in statistical analyses because their distribution is very heterogeneous, having zero value a large part of the time, and scarce rainfall episodes in large ranges of intensities and durations. We thus computed three values: (i) for each cluster of days, the total number of dry days (defined as days with zero rainfall), and for each day, (ii) the number of rainy hours (non-null hourly rain amount), and (iii) the

total amount of rain.

### 2.3.2    Thermally driven circulations

Thermally induced circulations in mountainous regions are local air motions induced by the heating of the air along mountain slopes. Close to the surface, air moves upward in the daytime (anabatic transport) and downward (katabatic transport) at night. Such transports may occur at various spatial and temporal scales : at the sacle of each single radiated slope, at the scale of

secondary and primary valleys, and at the scale of the mountain massif itself. (Whiteman, 2000).

Under clear sky and weak synoptic wind conditions, plain-to-mountain transport can be set off in the daytime at a regional scale (e.g., up to 100 km in the Bavarian Alpine foothills, Lugauer and Winkler, 2005). A closed circulation cell may form, with a return flow at altitude, oriented from the mountain to the plain. Such circulation and its impact on atmospheric composition has been specifically studied at the P2OA by Hulin et al. (2019). These authors propose three detection methods that will be

applied to our time-period. Technical details about the methods are available in their article. The main concepts are summarized here.

Method 1 aims at detecting the presence during the daytime of a return flow above the CRA by comparing the wind at 3000 m (that could be affected by the circulation) and at 5000 m (presumed to be unaffected). The main idea is to find in the interval 10-16 UTC (but not before and after) a significant enhancement of the southern component of the wind at 3000 m that would

be negligible at 5000 m.

Method 2 aims at detecting anabatic/catabatic surface breezes at the CRA, by considering the diurnal alternation in wind direction: north-easterly during daytime (11-14 UTC), and south-easterly at night (00-02 and 21-00 UTC). The latter two methods result in daily boolean flags which show whether or not a thermally driven circulation is detected for the current day.

Finally, Method 3 consists of ranking days of the dataset depending on the influence of anabatic transport on the water vapour

content measured in situ at PDM. This influence can be quantified using the amplitude of the diurnal cycle of specific humidity as a proxy. The larger this amplitude, the more efficient the anabatic transport of humid air from the valleys to PDM. This



method was originally designed by Griffiths et al. (2014) from radon measurements. However, radon data were not available for Hulin et al. (2019)'s period of study (2006-2015), so they alternatively used specific humidity as suggested by Griffiths et al. (2014) as an alternative. In our case, specific humidity and radon data are simultaneously available from late 2017 to the end of 2019, a period over which we checked that rankings based on both variables provided consistent results (not shown). In the following, we therefore only consider the ranking based on specific humidity, as humidity data were available at PDM for the whole timeframe of the study. Method 3 assigns a rank to each day according to the degree of anabatic influence: the day ranked 1 has the diurnal cycle with the largest amplitude; then the amplitude decreases as the rank increases, until vanishing at a threshold rank (day ranked 850th) after which no more diurnal cycle can be observed. So, the method allows to distinguish us anabatic days (ranked before the threshold) from non-anabatic days (ranked after). All details can again be found in Hulin et al. (2019).

### 2.3.3 Foehn

Jansing et al. (2022) define the generic term *foehn* as "downslope winds and windstorms in the lee of mountains [...] associated with a distinct warming and a decrease in relative humidity of the air on the lee side of the orographic barrier". On the northern side of a mountain barrier, which is the case at the P2OA, foehn situations (south foehns) require south-westerly to southerly synoptic flows.

Foehn occurrence and characteristics will be studied by means of two diagnostic tools. The first is the horizontal pressure difference across the Pyrenees. Foehn, which is in essence a cross-barrier wind, is typically associated with a pressure dipole across the mountain chain (Bessemoulin et al., 1993). In the case of a southerly foehn, there is therefore a positive pressure difference between the south and the north of the chain. This pressure drag increases with the intensity of the foehn (Lothon et al. (2003), Drobinski et al. (2007)). To compute this diagnostic tool, pressure data was needed from the Spanish side. We used mean sea level pressure data every 6 hours from the NCEP global reanalysis (https://rda.ucar.edu/datasets/ds083.2/#!description), taken at the nearest grid point from Monzon (130 km south of CRA). For pressure on the French side, we had two possibilities: the actual pressure measured at CRA (reduced to msl) vs. the pressure data from NCEP reanalysis at the closest grid point. Higher differences were found with the measured pressure (suggesting that NCEP reanalysis could underestimate the foehn intensity due to the smoother terrain in the model.), therefore we retained the measured pressure at CRA for the calculation.

The second diagnostic tool is based on the occurrence of mountain lee waves, as seen over the CRA with the VHF wind profiler. During a foehn event, the south to southwesterly flow generates mountain waves in the lee of the Pyrenees chain, which can be observed throughout the whole troposphere. Figure 3 shows an example of such a situation, as seen by the VHF wind profiler. This figure shows how the strong southerly flow (Fig. 3a) can be associated with large variations of vertical air velocity (Fig. 3b), a signature of the mountain lee waves. The intensity of vertical tropospheric oscillations is here quantified with the variance of vertical wind w at 2850 m a.s.l. computed over a running 6-hour interval, from the original 15-min VHF-profiler data. A data point is flagged as a lee-wave occurrence if the horizontal wind direction is between 150° and 250° and if the w variance exceeds $0.1 m^2 \ s^{-2}$. Corresponding time series of vertical velocity, variance and wind directions are displayed in Fig3(c) and (d) with the identification of the lee wave occurrence with this method. We considered as a foehn hour any hour



**Table 3.** Diagnostic tools used in the study. (See text for details.)

| Diagnostic tool | Unit | Timestep | Output type | Comments |
|---|---|---|---|---|
| **Vertical Structure** | | | | |
| Difference of potential temperature between PDM and CRA | K | Daily | Numerical | From in situ temperature and pressure data |
| Convective Boundary Layer Height | m | Hourly | Numerical | Estimation by Philibert (2023) from UHF wind profiler data |
| Sensible heat flux anomaly (H) | W m$^{-2}$ | Daily | Numerical | From high rate measurements of wind and temperature |
| Latent heat flux anomaly (LE) | W m$^{-2}$ | Daily | Numerical | From high rate measurements of wind and moisture |
| **Precipitation** | | | | |
| Dry day index | % | Daily | Boolean | |
| Cumulative rainfall amount | mm | Daily | Numerical | Dry days are excluded from the statistics |
| Number of rainy hours per day | # | Daily | Numerical | Dry days are excluded from the statistics |
| **Thermally driven circulations** | | | | |
| Method 1: Return flow above the CRA | | Daily | Boolean | From VHF wind profiler data at 3000 m and 5000 m a.s.l. |
| Method 2: Diurnal surface breeze at CRA | | Daily | Boolean | From surface wind data |
| Method 3: Detection of anabatic days | | Daily | Boolean | From specific humidity at PDM |
| **Foehn** | | | | |
| Pressure difference across the Pyrenees | hPa | 6 hours | Numerical | Mean sea level pressure near Monzon, Spain, from NCEP reanalyses |
| Foehn day index based on lee wave detection | | Daily | Boolean | From VHF wind profiler data |

having at least 50% of the 15-minute data points flagged as lee-wave. Finally, in order to extract a daily diagnostic, we consider as a foehn day any day containing at least 6 hours of foehn.

## 3 Meteorological regimes from the hierarchical clustering

### 3.1 Clustering implementation and cut of the clustering tree

A hierarchical clustering algorithm was applied (with options detailed in Section 2.2) to a collection of 1826 events (observation days from the 1st of January 2015 to the 31th of December 2019), each composed of the 23 variables listed in Table 2 (upper part). Gas and particle concentrations (Table 2, lower part) were not included in the list of variables driving the clustering, with







**Figure 3.** Time-height plots of the horizontal (a) and vertical (b) wind measured by the VHF wind profiler on 6-8 April 2018, as an illustration of a foehn event. In (a), an arrow toward the top indicates a southerly flow, an arrow towards the right indicates westerly flow. Times series of vertical wind component $w$ (c) and wind direction (d) at 2850 m a.s.l. for the same days are shown. In all four panels, vertical red lines represent the beginning and end of the detected foehn episode. In panel (c), the red curve represents the rooted variance (i.e. standard deviation) of $w$ over a 6-h running interval, while the red horizontal line represents the threshold for the lee wave detection ($\sqrt{0.1\,\mathrm{m}^2\,\mathrm{s}^{-2}} \approx 0.32\,\mathrm{m\,s}^{-1}$). Green circles identify the data points for which the variance criterion is met. In panel (d) the two horizontal red lines represent the two wind direction thresholds used in the detection method.





the aim of obtaining regimes based purely on the local meteorology. Nevertheless statistics on gas and particle variables were considered in each meteorological cluster, and will be presented in Section 5.

Our choice to cut the clustering process at the step with 6 clusters allowed us to have a minimum number of clusters while keeping the size of the largest cluster below 50% of the total number of observation days (1826). We thus obtained three major (i.e. highly-populated) clusters (containing 622, 720 and 415 days, thereafter clusters 1, 2 and 3, respectively) and three minor clusters (containing 20, 33 and 13 days, thereafter clusters 4, 5 and 6, respectively).

### 3.2   Analysis of the three major clusters

**3.2.1   Thermodynamic variables**

To explore the characteristics of the major clusters, we first summarized the statistical distribution of the main thermodynamic variables within each class by means of box-and-whisker plots (hereafter "boxplots") in Figure 4. It shows that whatever the variable or class, the distributions are centered on median values showing marked differences between clusters (even though the interquartile ranges overlap in most cases).

Considering the temperature anomaly at PDM, which represents the deviation from the expected seasonal value, Clusters 1 to 3 have median values of +2.5 K, -0.5 K, and -4.5 K, respectively (Fig. 4a). A similar hierarchy also appears in the temperature anomaly at the CRA (+2.5 K, -0.5 K and -3.0 K respectively, Fig. 4b), the pressure anomaly at both stations (Fig 4c for PDM, CRA not shown) and the solar (downward shortwave) irradiance (Fig. 4d). We also noticed a reversed pattern (i.e., increasing median values for Cluster 1 to Cluster 3) for relative humidity (Fig. 4e for CRA, PDM not shown) and cloud cover at the CRA

(median values of 15%, 65% and 70%, respectively, Fig. 4f).

    In brief, Cluster 1 contains warmer, drier, clear-sky and high-pressure days, suggesting anticyclonic fair-weather conditions; Cluster 3 contains colder, more humid, cloudier low-pressure days, suggesting disturbed weather; and Cluster 2 contains days of intermediate characteristics.

**3.2.2   Wind**

Hodographs of the synoptic wind from the VHF profiler (at 2850 m a.s.l. corresponding to the altitude of PDM, Fig. 5) show that in Cluster 3, the wind blows mostly from the northwest quadrant. In addition, the wind strength is above 10 m s$^{-1}$ a large part of the time, and exceeds 20 m s$^{-1}$ on some days, whereas there are much fewer strong wind days in Cluster 2, and almost none in Cluster 1.

    In western Europe, strong northwesterly winds are typical of disturbed weather (with low temperature, high cloud cover and

rainfall, e.g., Giuntoli et al., 2021). In clusters 1 and 2, the wind may blow from a larger variety of sectors, with a few days with southerly or northeasterly wind, but nevertheless,the majority of days with southwesterly to northwesterly wind. However, Cluster 2 also shows strong (10-20 m s$^{-1}$) north westerlies that are almost absent in Cluster 1. Thus, Cluster 2 contains several days with similar wind conditions as Cluster 3. Cluster 2 also shows frequent days with strong southerly to southwesterly wind, potentially corresponding to foehn conditions.





**Figure 4.** Boxplots of temperature anomaly (K) at PDM (a) and CRA (b), of pressure anomaly at PDM (hPa) (c), of downward shortwave irradiance anomaly (W m$^{-2}$) at CRA (d), of relative humidity anomaly (%) at CRA (e) and of cloud cover (%) above the CRA (f).





**Figure 5.** Hodographs of the daily mean wind vectors measured by the VHF profiler at 2850 m a.s.l., for the 6 clusters (a-f). Wind speed (radius) is in m s$^{-1}$. Blue points correspond to days flagged as foehn based on lee wave detection (details in section 2.3.3)

.



Studying the wind above CRA but at a level below the Pyrenean crest (1600 m a.s.l., Fig.6) provides further information. In Cluster 3, the wind is concentrated in a narrower sector (between 270 and 300°) than higher in the mid troposphere. A plausible explanation is that the synoptic northwesterly wind is locally channelled along the Pyrenees at 1600 m. This channelling effect can also be seen in Cluster 1 and 2 from the west, but also from the east in some cases. When the synoptic wind is from the southwest, air masses may not have sufficient kinetic energy to flow over the Pyrenees, and in this case, they flow around the

barrier, with possible channelling on the lee side. Clusters 1 and 2 also contains a few days with sustained southerly wind – presumably south foehn events.

Lastly, Figure 7 shows the hodographs of hourly surface wind at CRA for both night and day. Due to the proximity of the Pyrenees, thermally-induced circulations are expected on sunny days, with wind blowing from the plain to the mountain in the daytime (northerly sector) and conversely at night (southerly sector). As expected, this alternation of wind between day and

night is most visible in Cluster 1, but also in Cluster 2, and to a much lesser extent in Cluster 3. In Cluster 2, strong southerly wind is sometimes observed even in the daytime, again presumably corresponding to foehn events.

### 3.2.3 Seasonality

Figure 8 shows the occurrence frequency of days of each cluster within the four seasons. Clearly, for all clusters the frequencies deviate from an equal distribution (25% of days in each season). This demonstrates the fact that weather regimes have their own

seasonality. Cluster 1 has an excess (32%) of summer days, consistent with the main characteristics (fair weather anticyclonic days). In the same way, cluster 3 has 33% of winter days and only 14% of summer days which is consistent with disturbed weather being more frequent in winter and spring. Cluster 2 has a deficit (19%) of winter days, but no explanation is obvious to us.

### 3.2.4 Global portrait of the major clusters

To summarize the last three paragraphs, we attempted to depict the main characteristics of weather regimes emerging from the three major clusters.

- Cluster 1 is characterized by high-pressure clear-sky, warm, dry and weak-wind days, during which thermal surface breezes develop over the Pyrenean foothills. Cluster 1 will subsequently be referred to as the fair-weather cluster.

- Cluster 3 will be called the atmospheric disturbance cluster, as it is characterized by sustained northwesterly wind, with

cold, wet and cloudy conditions.

- Cluster 2 contains days characterized by intermediate values for most variables, having similarities with days in either Cluster 1 or 3. Thus, this cluster is much more difficult to portray. Cluster 2 also contains days with characteristics of foehn days that will be investigated later with specific diagnostic tools.





**Figure 6.** Hodographs of the daily mean wind vectors measured by the UHF profiler at 1600 m a.s.l., for the 6 clusters (a-f). Wind speed (radius) is in m s$^{-1}$.





**Figure 7.** Hodographs of hourly mean surface wind at the CRA (m s$^{-1}$) for clusters 1 to 3, separating night (23-02 UTC, right panels) and day (11-15 UTC, left panels).







**Figure 8.** Seasonal occurrence of days in each cluster (in % of all days in the cluster).



### 3.3 Analysis of the three minor clusters

**3.3.1 Winter windstorms (Cluster 4)**

Cluster 4 contains only 20 days but is characterized by extreme values for a several variables. It reveals wind patterns similar to Cluster 3, with northwesterlies in altitude (Fig. 5d) but channelled along the Pyrenees (thus from the west) below the crest level (Fig. 6d). However, unlike Cluster 3, Cluster 4 contains only strong wind days (daily averaged windspeed at 2850 m a.s.l. between 18 m s$^{-1}$ and 35 m s$^{-1}$, Fig.5d).

In addition, Cluster 4 has the densest cloud cover of all clusters (median above 70%, Fig. 4f), just above Cluster 3. However, these two clusters differ strikingly in terms of temperature anomalies (Figures 4a and b), Cluster 4 revealing positive anomalies (i.e. temperatures above the seasonal mean) at both PDM and CRA. The seasonality of Cluster 4 is remarkable, with more than 75% of winter days and none in summer. The positive temperature anomaly may be explained by the rapid advection of oceanic air to CRA which is, in winter, warmer than continental air.

We can therefore this Cluster 4 as a collection of winter windstorms.

**3.3.2 Foehn (clusters 5 and 6)**

Cluster 5 and 6 share similar wind characteristics, with southwesterly winds at 2850 a.s.l. (Fig. 5e-f) and southerly winds at 1600 m a.s.l. (Fig. 6e-f). The median temperature anomalies at PDM (Fig. 4a) are similar for both clusters and slightly positive (i.e. temperature a bit above the sesonal mean). Comparing Figure 4a and b shows that for clusters 1-4, the temperature
anomalies are very similar at PDM and CRA, which is expected given the small distance between the stations (28 km). Strikingly, this is not the case for Cluster 6 where we can see a much higher positive temperature anomaly at CRA (above +5°C, highest median value of all clusters) than at PDM (Fig. 4b). This is also true for Cluster 5, but to a lesser extent. Lastly, the relative humidity anomaly is negative for both clusters 5 and 6 (Fig.4e).

Southwesterly wind in altitude and southerly wind below the crests, in combination with warmer and drier air at CRA,
strongly suggest that cluster 5 and 6 correspond to foehn situations. We can speculate even further that in Cluster 6, CRA is more often reached by more intense foehn flows than in Cluster 5, as suggested by the higher positive temperature anomaly at this site. Note, however, that more frequent southerly wind was not observed, nor were there stronger southerly winds in Cluster 6 than in Cluster 5.

The attribution of foehn situations is also in line with the negative pressure anomalies at the PDM (Fig. 4c) because PDM
is downwind of the main Pyrenean crest (Fig. 1). However, this pressure anomaly could also be due, at least partly, to the fact that foehn events often precede the arrival of pressure lows from the Atlantic. The attribution is also consistent with the seasonality of Clusters 5 and 6 (Fig. 8), as the foehn is a phenomenon that mainly occurs in spring and autumn (according to studies conducted in Alpine regions, for example, Bouët (1972) and Richner and Gutermann (2007)). We found no literature reference on the foehn climatology in the Pyrenees). In the next section, using diagnostic tools specifically built to detect or
characterize foehn events, we will check whether these two clusters contain the most Foehn events.





| | | Cluster 1 : Fair weather | Cluster 2 : Mixed weather | Cluster 3 : Disturbed weather | Cluster 4 : Winter windstorms | Cluster 5 : Weak foehn events | Cluster 6 : Stronger foehn events |
|---|---|---|---|---|---|---|---|
| **Number of days** | (% of total) | 622 (34%) | 720 (40%) | 418 (23%) | 20 (1%) | 33 (2%) | 13 (1%) |
| **Weather** | CRA | P+ T+ RH- q- CC- | CC+ | RH+ P- T- CC+ | CC+ q+ T+ | T+ RH- | T+ RH- |
| | PDM | P+ T+ RH- q- | / | P- T-RH+ | / | P- | P- RH+ |
| **Wind direction** | PDM (surf.) | SW to NW | SW to NW | NW to NE | NW | SW | SW |
| | CRA (surf.) | W day SE night | W day SE night | W | W | S | S |
| | Synoptic (2850 m a.s.l.) | SW to NW | SW to NW | NW quadrant | NW | SW | SW |
| **Vertical structure** | | | | | | | |
| Median $\Delta\theta$ | | 11.7 | 10.8 | 8.9 | 8.8 | 8.2 | 6.4 |
| Mean $Z_i$ (m) | | 600 | 569 | 630 | 466 | 504 | 369 |
| % of days with defined $Z_i$ | | 67 | 58 | 45 | 20 | 53 | 36 |
| Latent heat flux (H) median anomaly | | 9.5 | 3.7 | 1.2 | -12.3 | -20.8 | -20.3 |
| Sensible heat flux (LE) median anomaly | | 5.4 | -4.6 | -0.9 | 5.4 | 8.8 | 7.0 |
| **Precipitation** | | | | | | | |
| % of rainy days | | 19 | 62 | 79 | 80 | 67 | 46 |
| Median daily rainfall (mm) | | 0.6 | 3 | 5.6 | 13.6 | 8.8 | 2.1 |
| Median number of rainy hours | | 1 | 4 | 7 | 14.5 | 5 | 3 |
| **Foehn** | | | | | | | |
| Median $\Delta P$ | | 0.4 | -1 | -2.2 | 0.0 | 2.4 | 4.0 |
| Fraction of hours with lee waves (%) | | 2.9 | 7.6 | 1.2 | 0 | 47.8 | 75 |
| Number of foehn days (% in the cluster) | | 32 (5%) | 95 (13%) | 6 (1%) | 0 (0%) | 23 (70%) | 12 (92%) |
| **Chemical variables** | $CO_2$ | - | / | + | / | / | / |
| | CO | - | / | + | / | / | / |
| | $CH_4$ | - | / | + | / | / | / |
| | $O_3$ | + | / | - | / | / | / |
| | Radon | / | + | + | - | - | - |
| | PART NB | + | / | / | - | - | - |

**Table 4.** Synthetic table of the results of the study. For the chemical variables and weather, + stands for "above the median of the whole dataset", - stands for "below the median of the whole dataset" and / stands for "close to the median". For example, in Cluster 1, for $CO_2$, - means that most of the $CO_2$ anomaly distribution is lower than the full dataset median. For the weather part, P stands for pressure, T for temperature, RH for relative humidity, q for specific humidity and CC for cloud cover.

## 4 Consideration of specific diagnostic tools

In this section the diagnostic tools defined in 2.3 will be applied to the data from each cluster, with the aim of refining the analyses conducted above, or to validate the conclusions. All the results are summarized in the synthetic Table 4 but detailed and commented below.



## 4.1 Atmosphere dynamics vertical structure

The first two diagnostic tools detailed in section 2.3.1 provide information about the vertical structure of the atmosphere. First, we can see in Fig. 9a that for the major clusters, the median difference of $\Delta\theta$ is the highest for Cluster 1, which indicates a more stable atmosphere than in Cluster 2 and 3, consistent with fair weather and anticyclonic conditions. Cluster 3 and 4 have both low $\Delta\theta$, suggesting a less stable troposphere, which is to be expected in disturbed weather. Finally, we notice that Clusters 5 and 6 also have the lowest $\Delta\theta$ among the 6 clusters: this will be discussed further in section 4.4.

H median anomaly is also the highest for Cluster 1 (Table 4) which is consistent with fair weather conditions as the ground receives a great deal of sunlight which is transferred into the atmosphere as heat flux. The presence of clouds and rain in clusters 2 and 3, is less favorable to surface heating and surface convective instability, leading to a smaller positive anomaly in H and a negative anomaly in LE. The anomaly of H becomes negative in Clusters 4, 5 and 6. In the situation of winter storms (Cluster 4) we saw earlier that the temperature anomaly is positive in this cluster due to advection of warmer oceanic air. Here the surface layer is usually stable or dynamically mixed, as clouds and rain prevent dry convection. Sensible heat flux is weak or even negative. In foehn situations, the anomaly of H is also negative because foehn brings warm air to the CRA, also leading to weaker or even negative sensible heat flux (as air gets even warmer than the ground). Conversely, LE anomaly is positive in foehn clusters. In fact, the drier and warmer ait of the foehn strongly favours the evaporation of soil moisture. Strong winds in cluster 4 also favor evaporation, even if this is less than in the dry foehn clusters. This could explain the smaller positive anomaly of LE in cluster 4. A positive anomaly of the same magnitude is found in Cluster 1, but is associated with the solar heating of the surface.

The distribution within clusters for the $Z_i$ estimation (CBL height) may be affected by how the estimation is made. To get a $Z_i$ estimation of good quality, a well formed CBL is needed capped by a more stable and laminar atmosphere, which is generally not the case during disturbed weather events. This induces a difference of data availability of $Z_i$ estimations between clusters. By comparing the availability of the UHF and of the estimation, we were able to compute the percentage of days, for each class, with the $Z_i$ not defined. This provides information about the proportion of disturbed days within each class. As expected, among the major clusters, Cluster 3 has the lowest number of available $Z_i$ estimations (44.7%, Table 4). However, the boxplot 9b shows that the median $Z_i$ are similar in the three major clusters, the difference being inferior to the UHF resolution (75 m). Cluster 4 has the lowest number of available $Z_i$ days (19.8%) which is consistent with the results of Cluster 3. Comparing the two foehn clusters, Cluster 6 has more unavailable days than Cluster 5 (63.8% and 47.1%). With the assumption made earlier that Cluster 6 contains stronger foehn events, we can assume that the downslope hot wind prevents CBL from forming, or tends to squeeze it near the ground, typically below 500 m a.s.l. causing a lower median $Z_i$ in Cluster 6. Smaller $Z_i$ in foehn clusters is also supported by the negative anomaly of H fluxes. With smaller buoyancy flux at the surface (due to the warm air), CBL growth is significantly reduced.





**Figure 9.** Boxplots of (a) the mean daily difference of potential temperature $\Delta\theta = \theta_{\mathrm{PDM}} - \theta_{\mathrm{CRA}}$, (b) the convective boundary layer $Z_i$ (from UHF-profiler data), (c) the daily rainfall (dry days excluded, in mm), and (d) of the number of rainy hours per day (dry days excluded). On top of panel (c) are written the percentage of rainy days in each cluster.





## 4.2 Precipitation

In this section, we discuss the occurrence of precipitation in the different clusters, based on the diagnostic variables defined in Section 2.3.1, starting with the fraction of dry days (days with no rain) (Table 4). As expected, the clusters with the most frequent rain occurrence are Clusters 3 and 4 (79% and 80% of rainy days respectively), and the least rainy is Cluster 1 (19% of rainy days).

In addition, the rainy days in Cluster 1 are characterized by short episodes (mostly less than 5 hours) and small daily amounts (Fig. 9c-d). Figure 9c reveals many outliers in clusters 1, 2 and 3, corresponding to large daily amounts of precipitation (not shown for clarity issues). In contrast, there are much fewer long-lasting episodes in those clusters (outliers in Fig. 9d). This suggests that the heaviest rainfalls correspond to convective storms. As expected the median rainfall and rain duration are the highest in Cluster 3 and 4 – the latter (winter windstorm days) being the one with the highest values.

Comparing the two foehn clusters, Cluster 5 contains more rainy days than Cluster 6, in line with less solar radiation (Fig 4d), more humidity (Fig 4e), and a wider range of cloud cover fractions (Fig 4f). This supports the idea that Cluster 6 is characterized by more intense foehn, as subsidence can commonly cause adiabatic heating (high temperature anomaly, Fig 4b), air drying, cloud evaporation and convection inhibition.

## 4.3 Thermally driven circulations

The 3 methods by Hulin et al. (2019) were designed to detect thermally-induced flows at different scales and locations: (1) the altitude return flow of the plain-to-mountain pumping, (2) the surface breeze at CRA and (3) local anabatic influence detected in situ at PDM (respectively referred to as Methods 1-3). These methods have been applied to our data set and the results analysed by clusters (Table 5).

Hulin et al. (2019) evidenced that for a given day in their database, the meteorological conditions did not always meet (or miss) the criteria of all 3 methods at the same time. They concluded that these different types of thermally-induced flows are not systematically concurrent and may occur independently from each other. This may explain why, in our case (Table 5), the percentages obtained for a given cluster differ between methods. Nevertheless, all the methods lead to a common hierarchy when clusters are compared to each other.

**Table 5.** Results of the 3 detection methods of thermally-induced circulations (detailed in Section 2.3.2).

| | | Cluster 1 | Cluster 2 | Cluster 3 | Cluster 4 | Cluster 5 | Cluster 6 |
|---|---|---|---|---|---|---|---|
| **Method 1 : Return flow above CRA** | (% of detected days) | 25 | 24 | 15 | 0 | 12 | 23 |
| **Method 2 : Surface breeze at CRA** | (% of detected days) | 60 | 31 | 2 | 0 | 49 | 39 |
| **Method 3 : Anabatic influence at PDM** | (% of detected days) | 62 | 46 | 40 | 10 | 33 | 15 |

Concerning the 3 major clusters, all 3 methods agree with the fact that Cluster 1 contains the most frequent anabatic days, and Cluster 3 the least. This is consistent with the main characteristics of the clusters depicted in Section 3, as thermally-driven circulations need a context of low synoptic wind and sufficient solar radiation to develop. These conditions are typical of





Cluster 1, partly occur in Cluster 2 and are rare in Cluster 3. We noticed that Method 2 (breeze at CRA) is the method that gives that largest discrepancies between Cluster 1 and 3 (60% and 2% respectively). This very low occurrence in Cluster 3 is
not surprising when considering that strong westerly winds are more frequent in this cluster than in the other two (Fig. 5 and 6), especially close to the surface at CRA (Fig. 7). Cluster 4, composed of winter windstorms, is also characterized by strong winds (Fig. 5-7), and in line with this, the first two methods detect no anabatic days at all, and method 3 reveals 90% of days with no anabatic influence at PDM.

Methods 1 and 3 give fewer occurrences overall of thermal flows for the foehn clusters (5-6) than Clusters 1-3 (Table 5) –
except for Cluster 6 where altitude return flows above CRA are found as frequently as in Clusters 1 and 2. In foehn conditions, sustained southerly or southwesterly wind at the altitude of the Pyrenean summits, in conjunction with subsidence on the lee side of the barrier, are not favorable for thermal flows to develop, or at least to reach such a high altitude, which explains the low occurrence rates in Table 5. The case of altitude return flows in Cluster 6 is hard to interpret physically, but this result may also be caused by the poor statistical representativity of Cluster 6 (only 13 days) and the risk of false detection of a return flow
by Method 1 if a short foehn event occurs in the middle of the day. Concerning Method 2, the relatively high occurrence of surface breeze at CRA for the foehn clusters (Table 5) could be seen as paradoxical in a synoptic context of sustained wind. However, unless foehn runs deep in the plain, the foothills are often sheltered from the foehn wind. Moreover, clear sky can easily develop in foehn conditions (Fig. 4d). In these conditions, thermal breezes can develop at the surface and be detected by Methods 2 and 3. The proportion is higher in Cluster 5 due to a deeper foehn in Cluster 6. In this situations, thermally-driven
winds in valleys and from plain to mountain are not incompatible with foehn.

Method 3 gives the same hierarchy between clusters 1-3 as the other methods but shows a significant percentage of anabatic days even for Cluster 3 (40%), which may be unexpected in disturbed weather conditions. To investigate the local anabatic influence at PDM, we plotted the mean diurnal cycle of radon for each cluster (Fig. 10). The daytime increase of radon is the clear signature of anabatic influence at a mountain-summit observatory (Griffiths et al., 2014). No evident diurnal cycle of
radon is visible in Figure 10 for the 3 minor clusters (4-6), as foehn and winter windstorms are not favorable conditions for thermal flows to develop close to the PDM summit as there is strong synoptic wind at altitude. Clusters 1-3, in contrast, exhibit clear diurnal cycles with a maximum of 14-15 UTC and a minimum at night. However, the cycle amplitude is above 1000 mBq m$^{-3}$ for Cluster 1, around 700 mBq m$^{-3}$ for Cluster 2, but is much lower (around 300 mBq m$^{-3}$) for Cluster 3. These amplitudes are in line with the hierarchy of anabatic day occurrence in clusters 1, 2 and 3. Even if the percentage of days with
detectable anabatic influence are close for clusters 2 and 3, the cycle amplitude in Cluster 2 is twice as large as in Cluster 3, suggesting that, in Cluster 3, the anabatic influence is notably less than in Cluster 2. This conclusion is eventually consistent with Cluster 3 containing days less favorable to local anabatic influence at PDM.

Figure 10 again supports the use of specific humidity for implementing Method 3 (as suggested by Griffiths et al., 2014), as the largest radon cycles coincide with the most anabatically-influenced days, as seen from the specific humidity point of view.
We can finally conclude that the occurrence of thermally-induced flows given by the three methods are globally consistent with expectations based on the main meteorological characteristics of the clusters (portrayed in Section 3), which are, or are not, favorable to thermal flow developments.



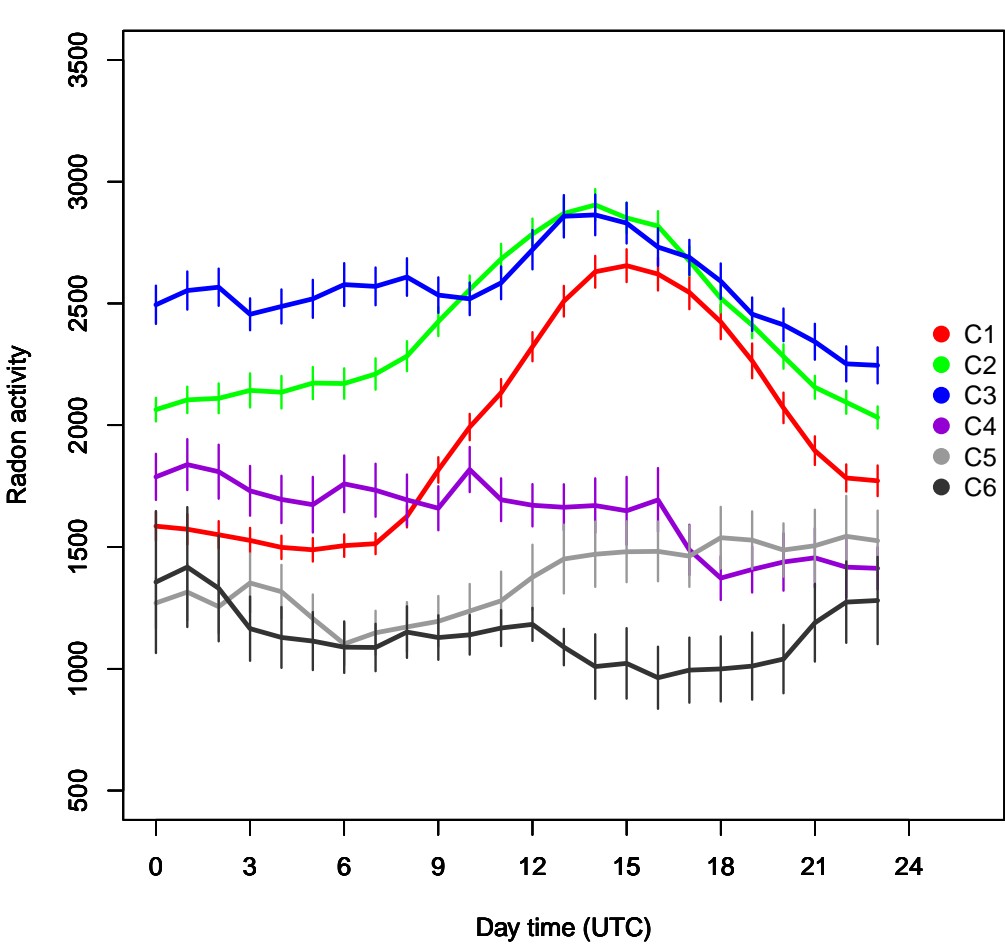

**Figure 10.** Mean diurnal variation of Radon (mBq m$^{-3}$) for each cluster (C1-6). Vertical segments represent the standard error.





### 4.4 Foehn

This part will focus on the diagnostic tools designed to characterize foehn events, starting with the pressure difference across
the Pyrenees ($\Delta P$), defined here as the upstream minus the downstream pressure. Foehn events should thus induce positive
$\Delta P$ values across the Pyrenees. However, due to the orientation of the main Pyrenean chain, a north-south pressure gradient
may also be associated with the westerly component of the geostrophic circulation. A crude estimation shows that 10 m s$^{-1}$
westerlies are driven by a 1 hPa difference over 100 km, which is the approximate distance between Monzon and the CRA.
Consequently, we will consider as the signature of foehn events pressure differences well above 1 hPa.

The median $\Delta P$ in clusters 5 and 6 are respectively of 2.4 and 4.0 hPa (Table 4), where Cluster 1 has a median $\Delta P$ of 0.4
hPa and all the 4 remaining clusters have negative $\Delta P$ (the lowest being Cluster 3 with -2.2 hPa). Negative differences found
for Clusters 2 and 3 can be explained by the prevalence of synoptic winds with a northerly component, which will induce a
reversed pressure dipole across the chain. Clusters 5 and 6 largely overcome the 1 hPa difference associated with geostrophic
westerlies which is consistent with the hypothesis of foehn events forming these clusters. The higher $\Delta P$ in Cluster 6 than in
Cluster 5 is also consistent with our interpretation of stronger foehn effect on the lee side (Section 3.3.2).

A second diagnostic of foehn events is the presence of lee waves above the CRA that can be detected with the tool described
in Section 2.3.3. This tool firstly gives the fraction of hourly timestamps flagged as foehn when lee waves are detected (Table
4). As expected, Clusters 5 and 6 have by far the largest fractions among the 6 clusters (48% and 75% respectively). Focusing
on the three major clusters, (1-3), most frequent foehn events are found in Cluster 2 (8%).

Then, a daily index was computed, a foehn day being considered as a day when a minimum of 6 hours had been flagged as
foehn. The total numbers and fractions of foehn days in the clusters are also presented in Table 4. The daily percentages are
found to be systematically higher than the hourly ones, due to the fact that short foehn events (6 hours) have the same weight
as episodes lasting a whole day in the daily flag. Interestingly, the number of foehn days in Cluster 2 (95) is in absolute greater
than the number of foehn days in Clusters 5 and 6 (35 days, adding both clusters). These events in Cluster 2 appear in Fig.
5b as the strongest winds in the SW quadrant. This means that the unsupervised clustering used here was not able to gather
all foehn days in specific clusters. This could partly be explained by the fact that those days also correspond to larger daily
rainfall relative to the rest of the days in Cluster 2 (not shown). Thus, they correspond to the situation (mentioned in Section 1)
of unstable southwesterly flows, with occurrence of storms. The very high proportion of foehn days in clusters 5 and 6 reveal
that the most intense foehn events have a characteristic signature in the 23 meteorological variables listed in Table 2. In Cluster
6, there is one day not flagged as foehn simply because the VHF data point is missing this day. For Cluster 5, 10 days are not
flagged as foehn with 4 because of missing data. The remaining 6 days have either wind too westerly to be in the scope of the
index or have a mean wind not strong enough to generate lee waves (Fig. 5e).





## 5   Impact on atmospheric composition

This section will investigate whether there are statistical differences in the atmospheric composition variables (Table 2, lower
list) between clusters, and will discuss what could explain those differences. A question of particular interest is the influence
of local and regional transport on atmospheric composition in the different weather types.

Boxplots of mole fraction anomaly of $CO_2$, CO, $CH_4$, $O_3$, of particle number concentration anomaly, and of radon activity
are displayed by clusters in Figure 11, and discussed in detail in the next sections. Before this, a general comment is that
the dispersion of values within a given cluster is usually quite large in many cases, revealing the complexity of physical and
chemical processes linking source regions and receptor, even in well-identified weather regimes. Nonetheless, when the clusters
are compared for a given variable, in most cases the distributions are sufficiently separated from each other to evidence true
statistical difference. As supplementary material we give the p-values of the t-test computed for each variable and for each
couple of clusters.

### 5.1   Radon as tracer of continental influence

As radon is constantly emitted from continental surfaces and is only subject to radioactive decay (half-life of 3.8 days), a high
radon activity at a mountain observatory like PDM reveals the transport of air influenced by the European surface (e.g. Griffiths
et al., 2014, for the Jungfraujoch). However it tells us little about the scale of the transport pattern. Transport of continental
radon-rich air may occur at a local scale, e.g. driven by anabatic flows in the close mountain area, or at a larger scale if the entire
regional low troposphere is subject to vertical mixing, e.g. in convective or frontal conditions. In the latter case, the synoptic
horizontal transport may also bring radon-rich air masses to PDM. On the contraty, in stable anticyclonic conditions, vertical
mixing tends to be inhibited, and the regional troposphere is expected to be radon-depleted at the altitude of the summits.

The median volume activity of radon (equivalent to the molar concentration[1]) is found to be lower, medium and higher in
the major Clusters 1, 2 and 3, respectively.

From the radon point of view, fair weather conditions prevailing in Cluster 1 are thus equivocal: on the one hand, atmo-
spheric stability should generate a regional context of low radon activity in the free troposphere around PDM; on the other
hand, anabatic influence should be favored in fair weather conditions (Fig. 10). The boxplots for radon in Fig. 11a resolve to
some extent this inconsistency: despite a wide distribution of radon values in Cluster 1, the median is the lowest among the
major clusters, suggesting that in the majority of cases, the daily mean radon concentrations at PDM seem to be under the
dominating influence of the regional context compared to daytime anabatic transport. Figure 10 further supports this statement.
The nighttime free-tropospheric radon background in Cluster 1 is much lower than in Clusters 2 and 3, and even the large
amplitude cycle is not sufficient to raise the radon activity above Clusters 2 and 3 at the time of the afternoon maximum. The
diurnal mean will thus clearly be lower in Cluster 1 (and the highest in Cluster 3).

---

[1]The proportionality factor being the radioactive disintegration constant: $\lambda_{222\,Rn} = 2.1\ 10^{-6}\ s^{-1}$.



**Figure 11.** Boxplots of (a) radon volumic activity (mBq m$^{-3}$), and of anomalies of (b) CO$_2$ ($\mu$mol mol$^{-1}$), (c) CH$_4$ (nmol mol$^{-1}$), (d) CO (nmol mol$^{-1}$), (e) O$_3$ (nmol mol$^{-1}$) and (f) particle number concentration (# cm$^{-3}$) – all variables measured in situ at PDM.



Looking at Cluster 4, the mean/median radon values are low (around 1300 mBq m$^{-3}$, Fig. 10 and 11a). We can speculate that during northwesterly windstorms, there is rapid advection of radon-poor oceanic air to PDM with limited mixing with the

continental boundary layer, but a backward particle dispersion analysis would be needed to support this.

Interestingly, the radon values are the lowest for the foehn clusters 5 and 6. Again a backtrajectography study is needed to explain this observation, but so far two assumptions (not mutually exclusive) can be made: (i) during their transport to PDM, the airmasses avoid flying over the western part of the Iberian peninsula, a hot spot of radon emissions in Europe (see e.g. the exhalation maps in Quérel et al., 2022). It should be noted that such an explanation could also be valid for the low values in

Cluster 4; (ii) during foehn episodes, PDM is located in the subsident part of the foehn wave in the lee of the Pyrenean crest, bringing radon-poor air from aloft to the station. In any case, the low radon values in foehn conditions clearly deserves more investigation.

## 5.2 Other gases and particles in the major clusters

The hierarchy found for radon between the three major clusters (1-3) (Fig. 11a) is also valid for the anomalies of $CO_2$, $CH_4$

and CO – namely: a negative median value in Cluster 1, near-zero in Cluster 2, and positive in Cluster 3 (Fig. 11b-d). Ozone and particle number anomalies display a reversed pattern – i.e. high values for Cluster 1 and low for Cluster 3 (Fig. 11e-f).

As $CO_2$, $CH_4$ and CO are primary pollutants mostly emitted from the surface (as is radon), the interpretations given for radon in Section 5.1 may be also valid for them. However, they have specific atmospheric sinks that should be considered.

Photosynthesis is the main sink of tropospheric $CO_2$ (Necki et al., 2003; Lin et al., 2017). The fair-weather days in Cluster

1 appeared to be warmer and to benefit from greater solar irradiance than the other major clusters (Fig. 4a, b and d). We can presume that under such conditions the photosynthetic activity was higher at the regional scale and could contribute to the observed $CO_2$ depletion. Note that the anabatic influence (favored in Cluster 1) can also contribute to the depletion of $CO_2$ daily mean, as the $CO_2$ diurnal cycle at PDM shows a daytime minimum caused by the local photosynthetic activity (Hulin et al., 2019).

For CO and $CH_4$, low anomalies in Cluster 1 could be due to a depletion in gas concentration at the regional scale due to enhanced oxidation by the hydroxyl radical (OH), produced in the troposphere by photolysis of water vapour (Seinfeld and Pandis, 2016). OH is a common sink of CO and $CH_4$ in the troposphere (Necki et al., 2003), especially in warm conditions with high solar irradiance. In Cluster 3 containing cloudy and cold days, in contrast, the atmospheric oxidative capacity is lower.

Inversely to radon, the mean tropospheric ozone profile shows a rapid increase with height in the lowest kilometers (Chevalier

et al., 2007; Petetin et al., 2018). The elements invoked to interpret the relative radon levels in Clusters 1-3 are thus again valid for ozone: enhanced atmospheric stability and free tropospheric influence may explain the higher ozone levels encountered in Cluster 1; enhanced mixing of the lower troposphere completed by regional horizontal transport to PDM may explain lower ozone levels in Cluster 3. Note also that anabatic transport brings ozone-depleted air to PDM (Hulin et al., 2019), but as for radon, this antagonistic effect does not obviously dominate the free tropospheric influence in Cluster 1. High ozone levels in

the troposphere can also be reinforced by enhanced photo-production in Cluster 1 (Fig. 4d). For the particle number anomaly,





without a deeper analysis of the aerosols at the PDM, we can only hypothesize that fair weather days enhance the production of small aerosols by photochemical reactions.

## 5.3 Other gases and particles in the minor clusters

Focusing on Cluster 4, the median anomalies are also negative for all gases and particles (Fig. 11b-f), accompanying low radon
levels (Fig. 11a). The assumption of rapid advection of baseline oceanic air to PDM, invoked for radon in Section 5.1, is also consistent for these other variables. Moreover, strong wind conditions favor atmospheric dispersion and dilution.

For the foehn clusters (5 and 6), we notice no influence on $CO_2$ (anomaly close to zero, Fig. 11b) but for the 5 other variables (including radon), the medians are negative and below the medians of the 3 major clusters (except $O_3$ in Cluster 5). The second assumption made in Section 5.1 to explain low radon levels that PDM is located in the foehn subsidence, seems to be applicable
to CO, $CH_4$ and particles as well, as for these variables, levels are expected lower in the free troposphere than in the boundary layer. As $CO_2$ has a much longer lifetime, it is well mixed in the troposphere. Positive $CO_2$ anomalies can only be observed very close to sources but rarely in the free troposphere. Therefore, it is not surprising to see no influence of foehn on $CO_2$. On the contrary, $O_3$ is expected at higher concentrations in subsident air masses. This is to some extent the case for Cluster 5 (with a large scatter, however) but not for Cluster 6. Further investigation would be needed on the origin and transport patterns of air
masses in Cluster 6 to explain those unexpected low ozone levels.

As foehn episodes correspond to south and southwesterly synoptic flows, they can be associated with dust transport from the Sahara, and in turn enhanced particle number concentration at PDM. Surprisingly, this is not found in the distributions of Clusters 5 and 6 (Fig. 11f), where even the 75-percentile values are low. Again, a trajectory analysis would be needed to determine the source region of air masses for these two clusters. Further checking is required to ascertain whether dust episodes
could be found in the foehn days characterized in Cluster 2 (see Section 4.4).

## 6 Summary and conclusions

The present study proposes a non-supervised classification of 5 years of a basic set meteorological observation data collected at both sites of the P2OA. CRA is the foothill site and PDM the mountain top site. Prior to this study, the diurnal and seasonal components of the time series, as well as the multi-year trends (when present), were filtered out in order to isolate the day-to-day
weather changes. The aim of this pre-processing and the subsequent classification is thus to form clusters of observation days with contrasting charateristics of the local meteorology, which could be related to synoptic weather regimes. Then, the statistical distributions of those data, but also of secondary diagnostic data tools, derived from the basic dataset or complementary observations, as well as atmospheric composition data at PDM, are analysed by clusters.

The used classification method used is hierarchical clustering computed with the complete linkage method. It resulted in 3
major clusters (numbered 1-3) and 3 minor clusters (4-6). All the results are summarized in Table 4, which helps us to draw a global portrait of each cluster, as follows:





- Cluster 1 (34% of the data collection) contains hot, dry clear-sky days, with weak to moderate wind in the free troposphere. Diagnostic tools designed to detect anabatic effects confirm that meteorological conditions in this cluster are the most favorable for the development of regional thermally-driven circulations and anabatic influence at PDM (Table 5). This cluster is the one with the highest proportion of summer days. This cluster was thus referred to as the fair-weather cluster. Under these conditions, low concentrations (relative to the seasonal mean) are found of radon, $CO_2$, $CH_4$ and CO, but high concentrations of ozone and total suspended particles.

- Cluster 3 (23%) contains cold, wet and cloudy conditions, with prevailing north westerlies in the free troposphere. It contains fewer summer days and more winter and spring days than an even distribution. This cluster is the rainiest among the three major clusters (79% of rainy days). Diagnostic tools designed to characterize the vertical structure of the lowest kilometres of the atmosphere indicate that Cluster 3 contains the least stable conditions among the three major clusters (lowest median $\Delta\theta$) and the least percentage of days with detectable boundary-layer top (only 45% of well-defined $Z_i$). This cluster was thus referred to as the atmospheric-disturbance cluster. In contrast to Cluster 1, high concentrations are found of radon, $CO_2$, $CH_4$ and CO, but low levels of ozone and total suspended particles.

- Cluster 2 (40%) is to some extent intermediate between Clusters 1 and 3, as it contains various types of situations, some similar to Cluster 1, and others to Cluster 3. Day occurrences are evenly distributed among seasons (fewer winter days, nonetheless). But notably, Cluster 2 contains 95 days detected as foehn days by the diagnostic tool specifically designed to detect lee-waves above CRA in case of sustained south-to-southwesterly synoptic wind (Section 4.4). Concentrations of all composition variables are found at intermediate levels compared to Clusters 1 and 3.

- Cluster 4 (1%) is composed of only 20 winter days but has very marked characteristics: strong northwesterly winds, the highest median daily rainfall (13.6 mm of rain, well above 5.6 mm for Cluster 3) and the highest median number of rainy hours of the collection (14.5 hrs, 7 hrs for Cluster 3). Rapid advection of oceanic air may explain the high temperature relative to the seasonal mean. This cluster was referred to as the winter-windstorm cluster. Concentrations found are low for all composition variables.

- Clusters 5 and 6 (2% and 1% respectively) are two clusters with clear characteristics of south foehn days: sustained south-to-southerly wind in the free troposphere, hot and dry air on the lee side, and a pressure dipole across the Pyrenees (highest median $\Delta P$ of all clusters). The lee-wave detection tool above CRA confirms 70% and 92% of days with lee waves in Clusters 5 and 6, respectively. In addition, the two clusters showed a significant difference with the other clusters in terms of heat flux anomalies, showing a positive anomaly on LE and a negative anomaly on H, consistent with advection of warm, dry air to the CRA. The two clusters differ in the intensity of the foehn effect on the lee side. The results suggest that foehn in Cluster 6 plunges deeper on the lee side than in Cluster 5, supported by the higher temperature anomaly at CRA (Fig.4b), higher median $\Delta P$ (Table 4), and more inhibition of the diurnal surface breeze at CRA (39% of surface breeze days in Cluster 6 but 49% in Cluster 5, Table 5). Days in Cluster 5 are mostly found in spring and autumn, and in autumn and winter in Cluster 6. While foehn has no obvious influence on $CO_2$ at PDM,



median concentrations are found to be low under foehn conditions for radon, $CH_4$, CO and particles, and high for ozone
        (but only in Cluster 5).

Weather-regime dependent concentrations have thus been found for the atmospheric species measured at PDM, and tentative
pieces of interpretation have been provided. Comparison of radon levels between Clusters 1 and 3 suggests that the regional
free-tropospheric background has a dominant influence on daily-averaged concentrations, prevailing over the daytime anabatic
influence. This may explain why, when photosynthesis and photochemistry are especially active at the regional scale (Cluster
1), concentrations are concurrently found to be low for $CO_2$, $CH_4$ and CO, and high for ozone and particles (assuming that
new particle formation is enhanced in that case, which is speculative and still to be confirmed by observations). In Cluster 4,
all six composition variables have negative anomalies, presumably due to the rapid advection of oceanic air to PDM. Finally,
for the foehn Clusters 5 and 6, we presume that foehn conditions bring mostly higher free tropospheric air to PDM, possibly
as the result of subsident transport. However, some results remain unexpected (less ozone in Cluster 6 than 5; no evidence
of transported Saharan dust), and tracking the air masses back to their source region remains necessary to give a consistent
interpretation for all species.

Applied to a basic set of preprocessed meteorological data, the hierarchical clustering provided here days ensembles where
the different variables are consistent with expected weather types. No contradictory information emerged, either between
variables within a given cluster, or between clusters. The diagnostic data products did not contradict, but on the contrary brought
further support to, the links established between the clusters and synoptic weather regimes, especially regarding the foehn
clusters 5 and 6, and allowed us to paint more consistent and comprehensive portraits of the clusters. Hence we can conclude
that hierarchical clustering of the local meteorological data may be a valid and simple approach to characterize the meteorology
of an atmospheric observatory – even in complex terrain – and its influence on the in-situ atmospheric composition.

This non-supervised approach, which requires no external data and no a-priori knowledge of the local meteorology, could
thus be easily carried out at other observation sites. A question that may arise, however, is whether the approach would be as
fruitful in sites with fewer meteorological observations, especially in the absence of observed wind profiles. Further study could
be to conduct sensitivity tests with our dataset in order to check the robustness of the obtained meteorological regimes when
part of the observations are removed (e.g., keeping near-surface wind data only, which are much more widespread than wind
profile data). Once proved robust for a few-year-dataset, one application of this approach could be to study the development of
the regimes over a long period in order to detect the impact of climate change on the occurrence and nature of weather regimes.

As a non-supervised approach however, hierarchical clustering may have limited ability to isolate a-priori known specific
phenomena. An illustration of this in our study, is foehn, because most lee-wave events were found in Cluster 2 which was not
specific to foehn. In order to make these foehn events fall into the specific foehn clusters 5 or 6, one could try to modify the
metrics used to compute distances between the data points, giving more weight to variables known to be linked to foehn (the
southerly wind component for example). One could also add the output data of the dedicated diagnostic tools (e.g. the binary
lee-wave index, or the cross-mountain pressure difference) to the list of variables driving the clustering. This would violate, at
least partly, the spirit of a non-supervised approach, but sensitivity tests would be nevertheless interesting to conduct. A purely
diagnostic and conditional approach would be complementary to the non-supervised approach.



Finally, adding another perspective, back-trajectography would bring valuable information about the origin of air masses in the different clusters (4, 5 and 6 in particular). This is needed in order to validate the interpretations proposed above and elucidate those results which remained unexplained.

## 6.1 Author contributions

J. Gueffier compiled the database, carried out the analyses and prepared the manuscript with contributions and reviews from
F. Gheusi, M. Lothon and V. Pont. A. Philibert designed the method of boundary-layer height detection from UHF data. The others co-authors contributed to data providing.

## 6.2 Competing interests

The contact author has declared that none of the authors has any competing interests.



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
