# Peer review of "Weather regimes and related atmospheric composition at a Pyrenean observatory characterized by hierarchical clustering of a 5-year data set"

_EGUsphere, 2023_

## Author Response (AR1)

We thank the reviewers for their positive and constructive reviews and for the comments which helped us to improve the manuscript. We answer below point-by-point to all general and specific comments.

**REVIEW 1**

The clustering method and the interpretation of the clusters are robust. However, the goal of the clustering (and subsequent interpretation of chemical characteristics of each cluster), whether it is for prediction (e.g. Neal et al. 2016) or for tracking the occurrences of certain weather types or others (e.g. Tso et al. 2020), is not very clear to me and needs to be highlighted better in the text.

We thank the reviewer for suggesting us these two very interesting article references, which can relevantly be discussed in our introductive section.

In Neal et al. 2016, clustering is used to define a limited number of meteorological types from an archive of model reanalyses over Europe. Then, these weather types are used to automatically sort great amounts of ensemble prediction model outputs, in order to facilitate human interpretation and predict the most probable meteorological scenario for the coming days.

The approach of Tso et al. 2020 is closer to the one presented in the present article. Among the variables in the database they mine, they distinguish « state variables », that serve to define a limited number of « states » by means of a clustering, from « observational variables » for which statistics are considered separately in the different states. Up to this point, their methodology is very similar to ours: a set of 23 meteorological variables observed locally at the P2OA will serve to define weather regimes; then the statistical distributions of concentrations of several atmospheric species will be compared in the different weather regimes. The final goals are different, however: while Tso et al. 2020 have a concern of data quality control on their observational data, and use extreme quantiles in the different states as criteria for flagging outliers, our main intent here is to characterize the main influences of meteorology on atmospheric composition at our observatory.

A reference to Tso et al. 2020 and a related discussion have been inserted in the text (lines 101-105 in the revised manuscript). This serves us further below in the revised text (lines 115-122) to clarify the goal of our clustering, by comparison to Tso et al.'.

The goal of Neal et al.'s study is further out of the scope of our study, and thus we found it hard to smoothly insert a related discussion in the flow of our text. Nevertheless, this study is a nice illustration of clustering based on model reanalysis pressure fields, and is now cited in line 97.

We also briefly exposed the main goal of the study, earlier in the abstract (lines 8-9) :

*« The objective of our study is to identify recurrent weather regimes at the P2OA, and link them to atmospheric composition patterns observed at the top of the PDM.»*

Specific comments:

L82: There has been more work on backward trajectory analysis and weather types (e.g. Tso et al. 2022)

Two other references have been added in Lines 85-87 (Tso et al 2022 and Gaudel et al 2015).

L91: this statement a bit long and confusing, consider rewriting

The references of the previous sentence were re-organised, and this statement refomulated (Lines 87-90)

Table5: may be better to present as pie/donut charts

We thank the reviewer for this suggestion.
In the present case, pie charts do not appear to us as optimal graphic representations, as every chart would represent a binary information (%age of selected vs. non-selected days within each cluster, for each method), and thus there would be as many charts as values given in the (former) Table 5. Alternatively, we chose to graphically display the data from Table 5 by bar charts in Figure 11, in which there is a group of 3 bars (representing the 3 methods) for each cluster. To complement this new figure, the numerical values of the former Table 5 have been integrated into the synthetic Table 4. These numerical values are especially useful for comparison with the past study by Hulin et al.

Section 6 summary and conclusion: Except the first paragraph, the other paragraphs (e.g. overall description of clusters, potential extensions, practicalities of hierarchical clustering) seem to fit better as a sub-section in the discussion, perhaps under the subheading "summary of findings and outlook". It will be good to keep the conclusion section and brief and high-level.

The final section of the article has been now re-organized in 3 parts as follows :
-Summary of the results.

-Discussion (the limitations of the methods and related perspectives).

- Conclusion, a short paragraph, giving high-level conclusions.

**REVIEW 2**

**MINOR COMMENTS**

Lines 124 to 128 – These lines are more appropriate in section 2 than in the introduction.

Effectively. The lines have been moved to the beginning of the 2.1.2 section (Lines 180-183).

Table 2 – Why the pressure at CRA is not used in the method?  Why w is only used at 2850 m agl and not closer to the surface?

Pressure data from CRA was not used to avoid redundancy with the PDM data. They both give a consistent pressure anomaly.

W at daily scale measured from the VHF at higher level gives access to subsidence and ascents at large scale. It was decided to consider the same level as the high level horizontal wind, i.e . at the height of the PDM. A measure of vertical velocity at surface would have a totally different meaning (close to zero), and is not relevant here.

*In our study, vertical velocity (w) from the VHF profiler is considered at 2850 m asl (as the horizontal wind), i.e. close to the altitude of PDM. At this height, and averaged at the daily time scale, w has non-negligible values (typically few ten cm/s) only in case of quasi-stationnary terrain-forced flows, typically foehn. In such case (as illustrated in Fig.3b), w is quasi uniform from the lower free troposphere (2 km asl) up to about 8 km asl. Hence, considering w at any other level in the free troposphere would be redundant with the information at 2850 m asl. At a lower level in the boundary layer (e.g. data from the UHF  or the 60m-tower), w is only significant at short time scale, and is driven by turbulence or thermal convection. Therefore daily-averaged values of w in the BL are almost zero and not relevant for our study,* and used at finer time scale in order to define the occurrence of mountain waves (foehn diagnostic). Including the daily-averaged  vertical velocity at 2850 m asl in the input variables of the clustering had the important consequence to separate  foehn situations into two specific clusters, namely 5 and 6, where w is markedly negative and positive, respectively. We indeed performed a sensitivity test (not shown in the article) where w was suppressed from the list of variables for the clustering. In this experiment, only one foehn cluster emerged, but included much less observation days than the sum of clusters 5 and 6. This brings two insights: first the existence of a quasi-stationary foehn-wave detectable at the daily time-scale above CRA, is a specific signature of foehn events (that would otherwise be put in a non-specific cluster like Cluster 2); second, the horizontal position of the foehn-wave (related with the sign of w) has an influence on the thermodynamic characteristics of the lower atmosphere at CRA and the penetrative character of the foehn influence on the lee side.

A discussion about the role of w in the clustering has been inserted in the revised manuscript version as well as a new figure in support of it (Fig.9). This discussion adresses the differences between cluster 5 and cluster 6 in foehn situations, based on both daily-averaged  vertical velocity at 2850 m asl, and the daytime surface wind at CRA.(lines  552-558) :

*« The information that clusters 5 and 6 are composed of days with a significant occurrence of lee waves also provides us a possible explanation for the difference seen in the boxplots of the vertical component of the wind (Fig.9c). We can speculate that the difference is due to a horizontal phase shift of the lee waves above the CRA. The boxplots show that Cluster 5 is in average associated with negative vertical velocity, while cluster 6 is associated to positive vertical velocity. This suggests that the positioning of the mountain wave may be different in both situations: during Cluster 6 cases, CRA is more frequently located in the ascending region, suggesting that the descending region may be closer to the mountain, allowing foehn penetration down to surface. More work (especially numerical modelling) is deserved on this specific topic. »*

Line 221 – NCEP is not a meteorological model. Please correct.

Correction to the description was made. NCEP is now referred as « NCEP reanalysis data » (line 294).

References to papers "in preparation". This fact should be checked with the journal, but I think the cited papers should be published before the publication of the present paper. Some examples are Lothon (2023), or Philibert (2023).

Philibert (2023) is now in revision at AMT, with online public discussion (https://amt.copernicus.org/preprints/amt-2023-95/) , hence now suitable for citation. All citation data have been updated in the bibliography (lines 850-852).

Lothon (2023) has not yet been submitted. If no suitable reference is still available by the end of the publication process of the present article, the url address to the P2OA web site will be given instead.

Table 3. Why "dry days are excluded from the statistics"? Please, clarify.

The two diagnostics of « cumulative rainfall amount » and « number of rainy hours per day » are used to characterize rainy days. Both values are zero for dry days, but as the latter are frequent, including dry days would considerably modify the shape of the boxplots in Fig.10(c) and (d), with an important loss of clarity. Instead, we preferred in these figures and related discussions to focus on the caracteristics of rainy days in each cluster.

We acknowledge that the mention « dry days excluded from the statistics » in Table 3 was confusing, and has been removed. The reason why this exclusion was done for Fig.10(c-d) is clarified in lines (lines 250-252).

Table 3 – Sensible and latent heat fluxes. Please, indicate the method used to calculate the fluxes (EC), as well as some characteristics (averaging window for example).

The method used to calculate the sensible and latent heat fluxes is now precised inLines 170-172 of the revised manuscript :

*« Heat fluxes are calculated on 30-min samples from high rate measurements (10 Hz) of temperature, wind and moisture, with the EddyPro® (Version 6.2.0). »*

Table 3 – Do the u and v components of the wind measured at CRA at 10 m come from the sonic measurements? (There is no other instrument specified in Table 2). If this is done like this and the hourly or daily averages are done first over the wind components, there is no risk for the wind direction calculation (circular variable).

u and v components of the wind measured at CRA at 10 m come from a standard meteorological station. In any cases in this study, averages were first performed over the wind components u and v, and directions and strengths of the obtained mean wind vector were calculated afterwards. Wind variables considered in the clustering are u and v, not the circular variable. Thus, there is no possible issues with the daily/hourly averages.

Line 261: Why the time interval during the night is not fixed? (00-02 and 21-00 UTC).

Those time intervals were defined by Hulin et al., because they referred to daily time intervals beginning at 00 UTC and ending at 23 :59. In such intervals, the nighttime period is sperated in two parts. While we also use such daily time intervals in our study, in the context of Line 261 it is simpler and less confusing to write the nighttime period as 21-02 UTC. The revised manuscript has been modified accordingly (Lines 268-269)

Line 287 – Please, provide the specific location of Monzon.

Geographical marker and coordinates of Monzon were added in Lines 295-296.

Figure 7. Why the format of these figures is different from the format of Figures 5 and 6? Please, consider unifying. Panel letters should also be added.

The format of figure 5-7 is now unified.

Lines 391-392 – Please, reword this sentence.

The sentence has been reworded (Lines 398-400), and  discussion and the new Figure 9 have been added.

*« This suggests further that foehn effect in Cluster 6 is more penetrative and affects more the surface on the lee side.  »*

Section 3.3.2 – clusters 5 and 6 are both considered as Foehn situations. Do you have an idea of their main differences? (more than those observed in Fig 4.). Foehn situations sometimes last for less than 1 full day, but the wind at the shown heights seems similar in both cases. Maybe the comparison of the wind at the surface for these two situations evidence some

interesting difference? Since the clusters do not include many cases, the authors could check some of the selected days to check their characteristics. This is a suggestion as future study.

The comparison of daily averages of the surface wind at the CRA shows no clear difference between Clusters 5 and 6. However, a clear difference is observed if focusing on the daytime, when the wind may come in some cases from the south (when the foehn wind dominates) or from the north (when the thermally-driven anabatic wind dominates). We have added in the new Figure 9 the hodographs of surface wind at CRA during the daytime, which better show the differences between cluster 5 and cluster 6. Cluster 6 exclusively shows souterly (foehn) wind during the daytime, while cluster 5 shows a mix of foehn and anabatic wind. Figure 9 also shows the daily-averaged  vertical velocity at 2850 m asl, which also reveals a marked difference between the two clusters.

The corresponding discussion has been added in lines 400-409.

Figure 9 – Most of the events in cluster 5 include some hours with rain, with an important amount of rain. 67% of cases are rainy. Is this a typical behaviour during Foehn cases? Please comment on it. This comment also applies to the last paragraph of section 4.2.

A paragraph was added at the end of Section 4.2 (lines 467-470) to comment the higher proportion of rainy days in Cluster 5 than in Cluster 6 :

« *Moreover, stormy situations over the Pyrenees are frequently associated with an unstable south-westerly synoptic wind. Thus, foehn situations are often followed by, or in line with, storms. This situation occurs generally in summer, which is consistent with Fig.8 where Cluster 5 contains summer days while Cluster 6 does not. This may explain the higher proportion of rainy days in Cluster 5.*»

Section 4.3 – Although the analysis of the three methods is very interesting and supported by the literature, the inclusion of 3 methods of detection here can be difficult to follow by the reader, in a paper in which many techniques are already used. I wonder if the authors could just focus on the methods that they consider the best. The paper is long and maybe this serves to add simplicity to it. This is a suggestion in case the authors consider that this simplifies the results, although there is an interesting discussion about their differences, but sometime difficult to prove.

We considered the suggestion of keeping only one or two methods in order to simplify the paper, but finally came to the conclusion to keep all three for the following reasons.

The three methods provide different points of view on thermally-driven circulations at the P2OA. Concerning the question of the influence on in situ compostion measurements at PDM, Method 3 is the most relevant here because it highlights best the influence of local circulations at the summit. Nevertheless, method 2 provides interesting insights on the concurrent occurrence of south foehn in altitude and anabatic breezes in the foothills. As

discussed above, this is obviously a differenciating feature of foehn events from clusters 5 and 6.

Finally, keeping the three methods (including Method 1) also allows us to compare the occurrence rates of thermal ciculations to those found by Hulin et al. for a different time period (2006-2015). The results for our all dataset (2015-2019) are found to be very consistent with theirs. To this goal, the percentage of days detected by all three methods for all the dataset has been added in Table 4 as well as in the new Fig. 11

Section 5. Some of the low concentrations of radon, CH4, CO2 can also be due to the enhanced mixing in summer, not only at CRA but also at Pic du Midi, i.e., due to the own PBL dynamics and higher PBL height.

The boundary layer in mountain environment is much more complex than over flat terrain. Even at a very local scale, it cannot be summarized as a turbulent or convective layer, in which atmospheric species initially concentrated near the ground would be diluted in air from aloft, thus causing a decrease of surface concentrations (as usually observed in flat terrain). De Wekker et al. (Boundary-Layer Meteorology, 113 : 249-271, 2004) showed that above mountains, there a considerable difference between the aerosol layer (observed by airborne lidar, and representative of the atmosphere influenced by the surface) and the convective boundary-layer, which exhibits a more terrain-following behaviour. In particular, anabatic flows on the slopes play a major role in advecting atmospheric species from the valleys. In general, when low-layer tracer concentrations are observed at a high-altitude site, the daytime development of anabatic flows and convection in the mountains does not tend to dilute the tracers, but in contrary to increase their concentrations, as well illustrated by radon (Fig.12 for Custers 1-3).

The O3 higher concentration can be related to the higher O3 formation under summer conditions (not discussed in section 5.2).

On average and in absolute value, ozone concentrations are indeed higher at PDM in summer, of course due to enhanced photoproduction in this season. In our study, however, we considered deseasonalized ozone data, i.e. anomalies with respect to the expected seasonal trend. But even in term of anomalies, our results evidence that fair weather days (Cluster 1) are favourable to positive ozone anomalies (despite daytime ozone depletion due to anabatic transport), as the regional free-tropospheric context is found to prevail. This is discussed in Section 5.2 (l.616-617).

Regarding the suspended particles, maybe more particles can reach these altitudes also because of the enhanced vertical mixing? Have the authors linked the values simply with the PBL height estimation?

Our results (Fig.13f) show on the contrary that daily-averaged particle concentrations are found lower under conditions of enhanced vertical mixing at the regional scale (Cluster 3) than in a stable atmosphere (Cluster 1). Our (speculative) interpretation is that photochemical

nucleation is favoured under fait weather conditions, and a high number of newly-formed small particles dominate the aerosol spectrum in such conditions. But it is also true that anabatic transport from the valleys can contribute to higher particle concentration in the afternoon (Hulin et al., 2019, their Fig.13e). Both effects are not mutually exclusive, as new particle formation may occur in both the free troposphere and the valleys. A more developed discussion has been inserted at the end of Section 5.2 (lines 621-631).

Concerning a possible link between the PBL height estimation (at CRA in our case) and particle concentrations at PDM, we do not think it is relevant to investigate this question. In the literature it is often stated that one moutain station is situated « above » (or « below ») the boundary-layer of the surrounding lowlands. Such a statement implicitely assumes that the lowland boudary-layer is viewed as a flat sea from which the mountain emerges like an island – or is submerged, depending on the boundary-layer height. As dicussed above, the boundary-layer processes in the mountains are very complex, and this simplified model is in most cases erroneous and misleading (especially for high and extended mountain massives). Even cases when the PBL top at CRA reaches or exceeds the altitude of PDM (and thus the lowland PBL could encompass the mountains) are actually extremely rare. For these reasons, the PBL height observed at  CRA is in almost no case relevant to discuss species concentrations measured at PDM.

Section 6. The main results are well summarized but there is also a full discussion in this section that maybe is more appropriate in an independent section.

Reviewer 1 also made comments on this section, which were consistent with this one. The final section has been re-organized into 3 parts : Summary, Discussion and Conclusion (see our response to Reviewer 1).

**Technical corrections**

All suggested technical corrections have been made.